# LatentKeypointGAN: Controlling GANs via Latent Keypoints

## Abstract

Generative adversarial networks (GANs) have attained photo-realistic quality in image generation. However, how to best control the image content remains an open challenge. We introduce LatentKeypointGAN, a two-stage GAN which is trained end-to-end on the classical GAN objective with internal conditioning on a set of space keypoints. These keypoints have associated appearance embeddings that respectively control the position and style of the generated objects and their parts. A major difficulty that we address with suitable network architectures and training schemes is disentangling the image into spatial and appearance factors without domain knowledge and supervision signals. We demonstrate that LatentKeypointGAN provides an interpretable latent space that can be used to re-arrange the generated images by re-positioning and exchanging keypoint embeddings, such as generating portraits by combining the eyes, and mouth from different images. In addition, the explicit generation of keypoints and matching images enables a new, GAN-based method for unsupervised keypoint detection.

## 1 Introduction

It is a long-standing goal to build generative models that depict the distribution of example images faithfully. While photo-realism is reached for well-constrained domains, such as faces, it remains challenging to make this image generation process interpretable and editable. Desired is a latent space that disentangles an image into parts and their appearances, which would allow a user to re-combine and re-imagine a generated face interactively and artistically. There are promising attempts (Lorenz et al., 2019) that use spatial image transformations, such as the thin plate spline (TPS) method, to create pairs of real and deformed images and impose an equivariance loss to discover keypoints and object appearance embeddings as the bottleneck of an autoencoder. Thereby, one can edit the image by moving the keypoints or modifying the appearance embedding. However, Li et al. (2020) points out that the learned keypoints may be biased and only meaningful for the artificially introduced image transformations, the TPS transformation require careful parameter tuning for each domain, and Xu et al. (2020b) shows that some geometry and appearance entanglement remains.

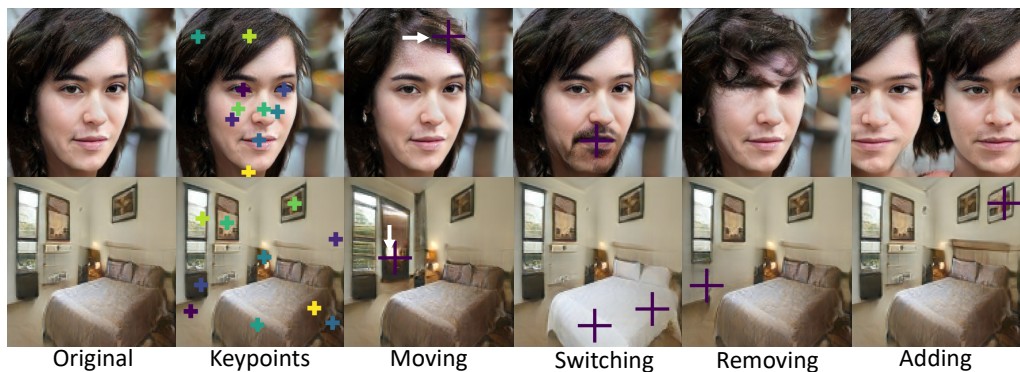

Figure 1: **LatentKeypointGAN** generates images (first column) with associated keypoints (second column), which enables local editing operations, such as moving, switching, and adding parts.

To avoid these shortcomings of existing autoencoder-based techniques, we propose an alternative GAN-based approach that does not rely on paired images and TPS transformation, thereby alleviating their domain-specific tuning and bias. Instead of an autoencoder, we use a GAN that includes keypoints as a latent embedding. Figure 1 shows how this *LatentKeypointGAN* enables our main goal of editing the output image by changing the keypoint position, adding or removing points, and exchanging associated appearance embedding locally. Although entirely unsupervised, the learned keypoints meaningfully align with the image landmarks, such as a keypoint linked to the nose when generating images of faces. As a byproduct, we can learn a separate keypoint detector on generated image-keypoint pairs for unsupervised keypoint detection and to quantify localization accuracy.

The difficulty of such unsupervised learning approaches lies in finding the right implicit bias to facilitate the desired disentanglement Locatello et al. (2019). LatentKeypointGAN is designed as a two-stage GAN architecture that is trained end-to-end on the standard GAN objective. In the first step, a generator network turns the input values sampled from a normal distribution into 2D keypoint locations and their associated encoding. We ensure with suitable layer connectivity that some of the encodings are correlated while others remain independent. These generated keypoints are then mapped to spatial heatmaps of increasing resolution. The heatmaps define the position of the keypoints and their support sets the influence range of their respective encodings. In the second step, a SPADE-like (Park et al., 2019) image generator turns these spatial encodings into a complete and realistic image.

We summarize our contributions below:

1. Development of a GAN-based framework for image manipulation and keypoint detection;
2. Design of a keypoint generator that models dependent and independent factors explicitly;
3. Empirical study comparing different editing methods in terms of perceptual quality;
4. A new methodology for keypoint detection that contests established autoencoder methods.

## 2 RELATED WORK

In the following, we discuss variants of deep generative models that learn to synthesize images from a collection of examples, focusing on methods modeling keypoints and those providing control on the image content.

**GANs** (Goodfellow et al., 2014) are trained to generate images from a distribution that resembles the training distributions. Recent approaches attain photo-realism for portrait images. Karras et al. (2018) train a progressive growth of the GAN's generator and discriminator. After that, Karras et al. (2019; 2020) integrate multiplicative neural network layers to build StyleGAN and StyleGAN2. StyleGAN gains some control on the generated high-resolution, high-quality faces, by adopting ideas from style transfer (Gatys et al., 2016; Huang & Belongie, 2017) and exchanging features of hidden layers between different samples. We build upon these network architectures and equip them with additional structure.

**Editing GANs globally.** More recently, efforts have been made on exploring the latent space of a pre-trained StyleGAN for image editing (Shen et al., 2020; Jahanian* et al., 2020). To allow editing real-world images, various encoders (Zhu et al., 2020a; Abdal et al., 2019; 2020; Guan et al., 2020; Wulff & Torralba, 2020; Richardson et al., 2020) have been trained to project images into the latent space of StyleGANs. These methods provide control over the image synthesis process, such as for changing age, pose, and gender. To enable rig-like controls over semantic face parameters that are interpretable in 3D, such as illumination, Tewari et al. (2020b;a); Ghosh et al. (2020); Deng et al. (2020) integrate 3D face models (Blanz & Vetter, 1999; Li et al., 2017) with GANs. Compared with these methods, our model focuses on detailed and local semantic controls. Instead of changing the face as a whole, our method is able to change a local patch without an obvious impact on other regions. Furthermore, our keypoints provide control handles for animation without manual rigging. Therefore, LatentKeypointGAN can be applied to many different objects and image domains.

**Editing GANs locally.** Instead of exchanging entire feature maps for style editing, local modifications are possible but require care to maintain consistent and artifact free synthesis. Collins et al. (2020) propose to cluster the middle-layer feature maps of StyleGAN and to use the embedding to

edit local appearance but require human supervision to select suitable clusters. Instead of using existing feature maps, Alharbi & Wonka (2020) and Kim et al. (2021) generate spatially-variable maps by structured noise generators and then swap the pixel embedding of the maps to edit the images. However, this requires to draw source and target regions by hand for every instance. By contrast, our model can *explicitly* change the pose by modifying keypoint locations that are consistent across images. Kwon & Ye (2021) manipulate attention maps to edit the pose and local shape, but they do not demonstrate local appearance editing. While enabling different forms of editing, all of the methods above require some form of pixel-level selection of regions at test time or manual cluster selection out of hundreds of candidates. In addition, they do not demonstrate on out-of-distribution editing, such as moving parts relative to each other and adding or removing parts, which requires stronger local disentanglement. Zhan et al. (2019) use a cycle consistency with a spatial transformation network to fuse a foreground object on top of a background reference. They can add a part but only if separate datasets with and without that part are available (e.g., with and without glasses). In sum, our keypoint-based editing is both more intuitive and enables additional editing capabilities.

**Conditional image synthesis** methods synthesize images that resemble a given reference input, such as category labels (Mirza & Osindero, 2014; Odena et al., 2017), text (Zhang et al., 2017; Reed et al., 2016), and layout (Zhao et al., 2019; Sun & Wu, 2019). A variant of the conditional GANs can do image-to-image translation (Isola et al., 2017; Huang et al., 2018; Liu et al., 2017; Zhu et al., 2017b;a; Wang et al., 2018; Park et al., 2019). These approaches aim to transfer images from one domain to another while preserving the image structure, such as mapping from day to night. Park et al. (2019) pioneered using spatially-adaptive normalization to transfer segmentation masks to images, which we borrow and adapt to be conditioned on landmark position. To control individual aspects of faces, such as changing eye or nose shape, recent works condition on segmentation masks (Lee et al., 2020; Yu et al., 2020; Richardson et al., 2020; Zhu et al., 2020b; Tan et al., 2020), rough sketches (Tseng et al., 2020; Chen et al., 2020), landmarks (Yang et al., 2019), or word labels such as with or w/o glasses (Cheng et al., 2020a), of faces as input. Hong et al. (2018) can superimpose a new object without requiring an exact mask, but still require one at training time. Compared with these methods, our model does not take any kind of supervision or other conditions at training time. It is trained in a totally unsupervised manner. Still, our method allows the landmarks to learn a meaningful location and semantic embedding that can be controlled at test time.

**Unsupervised landmark discovery** approaches aim to detect the landmarks from images without supervision. Most works train two-branch autoencoders, where shape and appearance are disentangled by training on pairs of images where one of the two is matching while the other factor varies. The essence of these methods is to compare the pair of images to discover disentangled landmarks. These pairs can stem from different views (Suwajanakorn et al., 2018; Rhodin et al., 2019), and frames of the same video (Siarohin et al., 2019; Kulkarni et al., 2019; Minderer et al., 2019). However, this additional motion information is not always available or is difficult to capture. Existing unsupervised methods trained on single images create pairs from spatial deformation and color shifts of a source image (Xing et al., 2018; Shu et al., 2018; Thewlis et al., 2019; Li et al., 2020; Cheng et al., 2020b; Dundar et al., 2020). However, the parameters of augmentation strategies, such as the commonly used thin plate spline deformation model (Thewlis et al., 2017; Zhang et al., 2018; Jakab et al., 2018; Lorenz et al., 2019), are difficult to calibrate and the learned keypoints may be biased to be only meaningful for the transformation (Li et al., 2020). Besides, the appearance information may be hidden in keypoint location (Xu et al., 2020b). The cycle consistency-based methods (Wu et al., 2019; Xu et al., 2020b) cannot provide local editing ability. We show that our generator can well disentangle the appearance and keypoint location and control the images locally. Furthermore, the underlying idea of these methods, no matter whether they depend on multi-views, videos, or thin plate splines, is creating pairs of images to compare to find the clue of equivariant keypoints in comparison. On the other hand, our method does not need any paired correspondences. It is totally based on generation. Our GAN approach poses a viable alternative by training a keypoint detector on synthetic examples generated by LatentKeypointGAN.

## 3 LATENTKEYPOINTGAN METHOD AND METHODOLOGY

The unsupervised spatial disentanglement of image content that we target has been shown to be impossible without an implicit bias imposed by the network architecture (Locatello et al., 2019). To this end, our proposed LatentKeypointGAN architecture encodes position explicitly and appearance

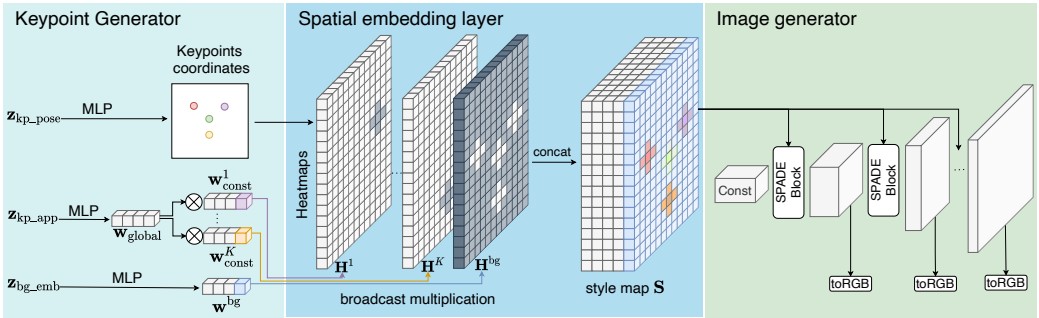

Figure 2: **Overview.** Starting from noise $\mathbf{z}$, LatentKeypointGAN generates keypoint coordinates, $\mathbf{k}$ and their embeddings $\mathbf{w}$. These are turned into feature maps that are localized around the keypoints, forming conditional maps for the image generation via SPADE blocks. The images are generated by *toRGB* blocks. At test time, the position and embedding of keypoints can be edited separately.

locally with two carefully designed sub-networks. First, the keypoint generator, $\mathcal{K}$, defines the spatial arrangement of parts and their embedding. Subsequently, a spatial embedding layer, $\mathcal{S}$, turns these sparse estimates into dense feature maps, and the image generator $\mathcal{G}$ up-scales these into a high-resolution image. Figure 2 shows the entire architecture. In the following, we explain architectural details and how we can train these networks end-to-end on a standard GAN objective, without any supervision on keypoints or masks. At inference time, the latent keypoints allow one to author the keypoint location and appearance interactively. Moreover, generated keypoint-image pairs can be used to train an independent keypoint detector $\mathcal{E}$, which enables unsupervised keypoint localization.

## 3.1 KEYPOINT GENERATOR

The keypoint generator, $\mathcal{K}$, learns the embeddings and spatial arrangement of image parts, such as eyes, nose, and mouth for describing a portrait image. It takes three Gaussian noise vectors as input $\mathbf{z}_{\text{kp\_pose}}, \mathbf{z}_{\text{kp\_app}}, \mathbf{z}_{\text{bg\_emb}} \sim \mathcal{N}(\mathbf{0}^{D_{\text{noise}}}, \mathbf{I}^{D_{\text{noise}} \times D_{\text{noise}}})$, where $D_{\text{noise}}$ is the dimension. Each vector is passed through a three-layer MLP to respectively generate the $K$ keypoint coordinates $\mathbf{k}^j \in [-1, 1]^2, j = 1, ..., K$, a global style vector $\mathbf{w}_{\text{global}} \in \mathbb{R}^{D_{\text{embed}}}$, and a background embedding $\mathbf{w}_{\text{bg}}$. Here $K$ is a pre-defined hyperparameter. Special is our keypoint embedding $\mathbf{w}^j$ that combines local and global factors,

$$\mathbf{w}^j = \mathbf{w}_{\text{global}} \otimes \mathbf{w}^j_{\text{const}}, \tag{1}$$

with the elementwise product $\otimes$. The constant embedding $\mathbf{w}^j_{\text{const}} \in \mathbb{R}^{D_{\text{embed}}}$ is designed to encode the keypoint semantics, e.g., left or right eye. They are updated during the training but fixed during testing. The global style vector $\mathbf{w}_{\text{global}} \in \mathbb{R}^{D_{\text{embed}}}$ can be regarded as a noise vector with learned distribution; it ensures that a different appearance is drawn for every generated image. We show in Appendix F that this factorization into dependent and independent factors is crucial and improves on straightforward variants that directly use an MLP. For instance, it facilitates learning the correlation of the two eyes of a face while still covering the wide variety of eye variations.

## 3.2 SPATIAL EMBEDDING LAYER

With keypoint coordinates and embeddings generated, we now turn these point-wise estimates into discrete style maps that are further processed by the convolutional generator. The style map $\mathbf{S}^j \in \mathbb{R}^{D_{\text{embed}} \times H \times W}$ for each keypoint $j$ is generated by broadcasting keypoint embedding $\mathbf{w}^j \in \mathbb{R}^{D_{\text{embed}}}$ at each pixel of the heatmap $\mathbf{H}^j \in \mathbb{R}^{H \times W}$ and rescaling $\mathbf{w}^j$ by the heatmap value $\mathbf{H}^j(\mathbf{p})$,

$$\mathbf{S}^j(\mathbf{p}) = \mathbf{H}^j(\mathbf{p})\mathbf{w}^j, \quad \text{where} \quad \mathbf{H}^j(\mathbf{p}) = \exp\left(-\|\mathbf{p} - \mathbf{k}^j\|_2^2 / \tau\right) \tag{2}$$

has Gaussian shape, is centered at the keypoint location $\mathbf{k}^j$, and $\tau$ controls the influence range. We also define a heatmap $\mathbf{H}^{\text{bg}}$ for the background as the negative of all keypoint maps, $\mathbf{H}^{\text{bg}}(\mathbf{p}) = 1 - \max_j \{\mathbf{H}^j(\mathbf{p})\}_{j=1}^K$. The background heatmap is multiplied with the independent noise vector $\mathbf{w}^{\text{bg}}$ generated from $\mathbf{z}_{\text{bg\_emb}}$ instead of keypoint embedding, but treated equally otherwise. Then we

concatenate all $K$ keypoint style maps $\{\mathbf{S}^j\}_{j=1}^K$ and the background style map $\mathbf{S}^{\text{bg}}$ in the channel dimension to obtain the style map $\mathbf{S} \in \mathbb{R}^{D_{\text{embed}} \times H \times W \times K+1}$. How these style maps are used as conditional variables at different levels of the image generator is explained in the next section.

## 3.3 IMAGE GENERATOR

The image generator $\mathcal{G}$ follows the progressively growing architecture of StyleGAN (Karras et al., 2019), combined with the idea of spatial normalization from SPADE (Park et al., 2019), which was designed to generate images conditioned on segmentation masks. Please refer to Appendix E for additional details. In the following, we explain how we replace the role of manually annotated segmentation masks in the original with learned keypoints. Our generator starts from a learned $4 \times 4 \times 512$ constant matrix and keeps applying convolutions and upsampling to obtain feature maps of increasing resolution. Following SPADE (Park et al., 2019), the original BatchNorm (Ioffe & Szegedy, 2015) layers are replaced with spatial adaptive normalization to control the content. By contrast to SPADE, we do not condition on annotated segmentation masks but instead on the learned feature maps introduced in Section 3.1, and we do not use the residual links (He et al., 2016) because we found it detrimental in combination with progressive training. The generator increases resolution layer-by-layer with multiple adaptive normalization layers, requiring differently-sized feature maps. To create feature maps that match the respective spatial resolution of the generator, we apply Equation 2 multiple times.

## 3.4 LOSS FUNCTIONS

**Adversarial losses.** We found it crucial for LatentKeypointGAN to use the non-saturating loss (Goodfellow et al., 2014),

$$\mathcal{L}(\mathcal{G})_{\text{GAN}} = \mathbb{E}_{\mathbf{z} \sim \mathcal{N}} \log(\exp(-\mathcal{D}(\mathcal{G}(\mathbf{z}))) + 1) \tag{3}$$

for the generator, and logistic loss,

$$\mathcal{L}(\mathcal{D})_{\text{GAN}} = \mathbb{E}_{\mathbf{z} \sim \mathcal{N}} \log(\exp(\mathcal{D}(\mathcal{G}(\mathbf{z}))) + 1) + \mathbb{E}_{\mathbf{x} \sim p_{\text{data}}} \log(\exp(-\mathcal{D}(\mathbf{x})) + 1) \tag{4}$$

for the discriminator, with gradient penalty (Mescheder et al., 2018) applied only on real data,

$$\mathcal{L}(\mathcal{D})_{\text{gp}} = \mathbb{E}_{\mathbf{x} \sim p_{\text{data}}} \nabla \mathcal{D}(\mathbf{x}). \tag{5}$$

Other losses such as hinge loss Park et al. (2019) failed.

**Background loss.** To further disentangle the background and the keypoints, and stabilize the keypoint location, we introduce a background penalty,

$$\mathcal{L}(\mathcal{G})_{\text{bg}} = \mathbb{E}_{\mathbf{z}_1, \mathbf{z}_2}[(1 - \mathbf{H}_1^{\text{bg}}) \otimes \mathcal{G}(\mathbf{z}_1) - (1 - \mathbf{H}_2^{\text{bg}}) \otimes \mathcal{G}(\mathbf{z}_2)], \tag{6}$$

where $\mathbf{z}_1$ and $\mathbf{z}_2$ share the same keypoint location and appearance input noise, and only differ at the background noise input. The $\mathbf{H}_1$ and $\mathbf{H}_2$ are the background heatmaps generated by $\mathbf{z}_1$ and $\mathbf{z}_2$. With this penalty, we expect the keypoint location and appearance do not change with the background. The final loss for the discriminator and the generator are, respectively,

$$\mathcal{L}(\mathcal{D}) = \mathcal{L}(\mathcal{D})_{\text{GAN}} + \lambda_{\text{gp}}\mathcal{L}(\mathcal{D})_{\text{gp}}, \quad \text{and} \quad \mathcal{L}(\mathcal{G}) = \mathcal{L}(\mathcal{G})_{\text{GAN}} + \lambda_{\text{bg}}\mathcal{L}(\mathcal{G})_{\text{bg}}. \tag{7}$$

## 4 EXPERIMENTS

We evaluate the improved quality and editing operations that our unsupervised LatentKeypointGAN approach for learning disentangled representations brings about. We show the types of edits in Section 4.3 and quantitatively and qualitatively evaluate our improved editing quality compared with existing unsupervised and supervised methods in Section 4.4 and appendix, respectively. To test the keypoint consistency, that keypoints stick to a particular object part, we compare to existing autoencoder frameworks for unsupervised landmark detection in Section 4.7. Even though not our core focus, we show that by further tailoring the proposed architecture for precise localization, this new GAN method reaches state-of-the-art accuracy on a keypoint detection benchmark.

Since our main goal of enhanced editing quality is hard to quantify, we also conducted a user study to further validate these improvements. The appendix contains exhaustive ablation tests on network architecture, hyperparameters, and training strategies to quantify the contributions of each individual component of our approach. The supplemental material contains additional video results.

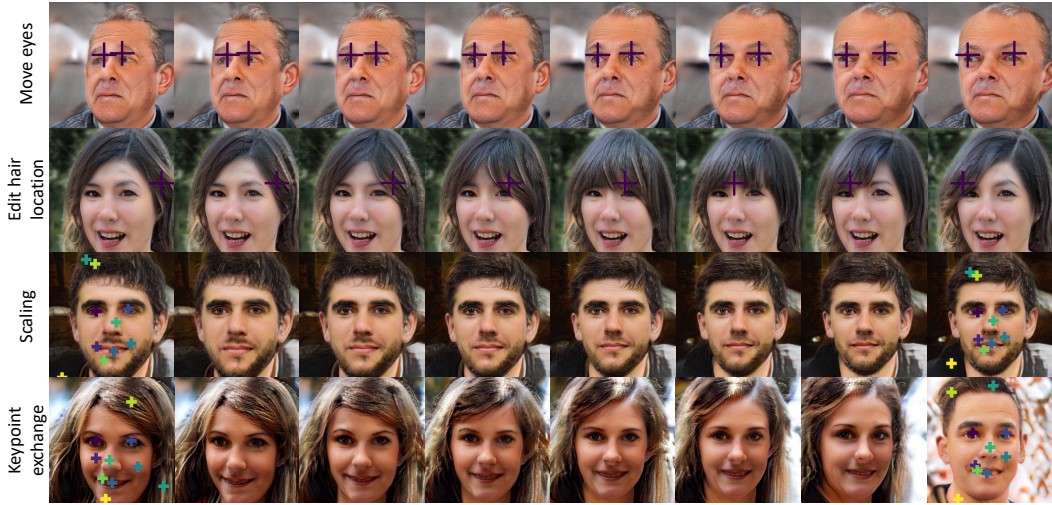

Figure 3: **Location and scale editing.** The first column is the source and the last the target. The images in-between are the result of the following operations. **First row:** pushing the eye keypoint distance from 0.8x to 1.2x. Note that the marked eye keypoints in this row are slightly shifted upward for better visualization. **Second row:** interpolating the hair keypoint to move the fringe from right to left. **Third row:** scaling the keypoint location and, therefore, the face from 1.15x to 0.85x. **Fourth row:** interpolating all keypoint locations, to rotate the head to the target orientation.

## 4.1 DATASETS

**CelebA** (Liu et al., 2015) contains 200k celebrity faces. We use this dataset to test our model's ability to discover the keypoints unsupervised. Following Thewlis et al. (2017), we use three subsets of this dataset: CelebA training set without MAFL (160k images), MAFL training set (19k images), MAFL test set (1k images). MAFL (Zhang et al., 2014) is a subset of CelebA. More details can be found in Section 4.7.

**CelebA-HQ** (Liu et al., 2015) contains 30k celebrity face images selected from CelebA. Following Zhu et al. (2020b), we use the first 28k images as training set and the rest 2k images as testing set.

**FlickrFaces-HQ (FFHQ)** (Karras et al., 2019) consists of 70k high-quality portrait images, with more variation than CelebA (Liu et al., 2015). Therefore, we use this dataset to test our model's ability to disentangle the local representations of images.

**BBC Pose** (Charles et al., 2013) consists of 20 videos of different sign-language signers with various backgrounds. We use this dataset to test our model's ability to edit human appearance.

**LSUN bedroom** (Yu et al., 2015) consists of more than 3 million images of bedrooms, with very diverse spatial structure. We use this dataset to test our model's generalization ability to edit entire indoor scenes.

**Hyperparameters.** We use $512 \times 512$ images for face editing on FFHQ, $256 \times 256$ for FID calculation, comparison with SEAN (Zhu et al., 2020b) on CelebA-HQ, and human pose experiments on BBCPose, and $128 \times 128$ for experiments on LSUN Bedroom, and comparison with unsupervised keypoint based methods on CelebA. Unless specified otherwise, we set $\tau = 0.01$ and use 10 keypoints. For editing experiments on FFHQ and detection experiments on CelebA, we augment by randomly cropping the training images to a size of 70-100%.

## 4.2 INTERACTIVE EDITING

We show the capabilities of LatentKeypointGAN by changing the keypoint locations and exchanging the keypoint embeddings between different images. As shown in Figure 3, we can thereby

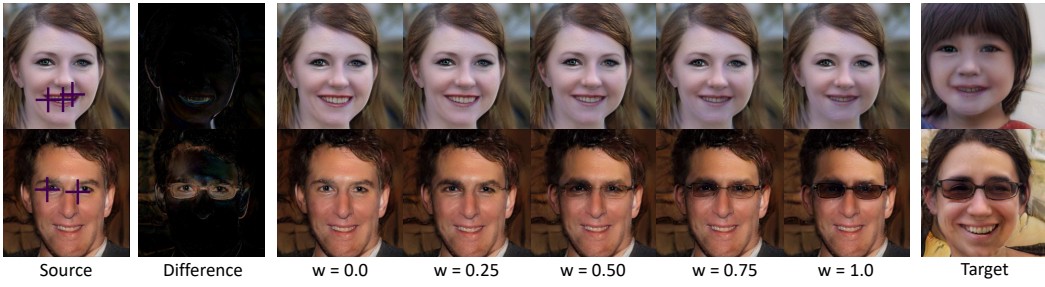

| Source | Difference | w = 0.0 | w = 0.25 | w = 0.50 | w = 0.75 | w = 1.0 | Target |

Figure 5: **Disentangled keypoint embeddings** on FFHQ. The leftmost images are the source and the rightmost images are the target. The cross landmarks on the first column denote the parts to be changed. The second column shows the difference between the original image and the changed image. The third to the second to last columns show the interpolation between the original image and the target image.

edit the face direction, face size, and individual key parts by changing the keypoint locations. If only a subset of the keypoint embeddings is changed, the other parts are not significantly affected. Figure 5 shows a heatmap of the area of interest. Since the GAN learns to generate a coherent face from its parts, global effects remain mostly unchanged, for instance, hairstyle and color. We discuss in Section 5 to which degree this is desired and what limitations remain.

Additional ablation studies, editing, and interpolation examples are shown in Appendix C, Appendix F, and the supplemental video.

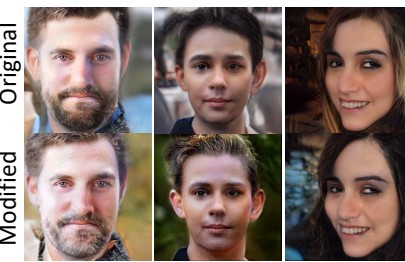

Figure 4: **Disentangled Background.** The background is changed while the faces are fixed.

### 4.3 DISENTANGLED REPRESENTATIONS

We first analyze how background and keypoint embeddings are disentangled and used for editing portrait images. Further examples are analyzed in the user study.

**Disentangled keypoint embeddings.** Figure 5 shows editing operations of independent facial regions. We fix the background noise, $z_{bg\_emb}$, and change some of the keypoint embeddings. This enables exchanging of eyes, mouth, or nose between persons. Figure 5 includes heatmaps that visualize the difference between original and interpolated images. Their local activation highlights the spatial disentanglement of individual keypoint features.

**Disentangled background.** Figure 4 shows a faithful change of backgrounds while keeping the face fixed. To this end, we fix the keypoint noise $z_{kp\_pose}$, $z_{kp\_app}$, and change only the background noise input, $z_{bg\_emb}$. The local change in the three diverse examples shows that the background and keypoint encodings are disentangled well. The illumination and hair color is learned to be part of the background, which makes sense as a global feature cannot be attributed to individual keypoints.

### 4.4 IMAGE QUALITY

We test our image quality on CelebA-HQ and compare it with an unsupervised autoencoder and segmentation mask-conditioned GANs, which require labeled training images. This comparison is biased in that the mask-conditioned GANs use the masks from the test set as input, while ours is unconditional. To make an as fair as possible comparison, we add a version that uses a self-supervised keypoint detector (see Sec. 4.7) to condition the latent keypoint locations on the same test images as the baselines do. Scores are computed by using the Pytorch FID calculator (Seitzer, 2020). We followed Zhu et al. (2020b) to use the first 28k images as the training set and the last 2k images as the test set. We list the results in Table 1. Our approach attains better FID scores than those GANs conditioning on segmentation masks. It gets close to SEAN (Zhu et al., 2020b),

| Method | Conditioned on | FID score ↓ |
|---|---|---|
| Pix2PixHD (Wang et al., 2018) | segmentation masks | 23.69 |
| SPADE (Park et al., 2019) | segmentation masks | 22.43 |
| SEAN (Zhu et al., 2020b) | segmentation masks & image | 17.66 |
| Ours | keypoints (unsupervised, see Sec. 4.7) | 21.71 |
| Ours (without detection) | none (unsupervised) | 20.12 |

Table 1: **Image quality with respect to supervision type** on CelebA-HQ. Our FID scores improve on mask-based solutions while providing similar editing capabilities and being unsupervised.

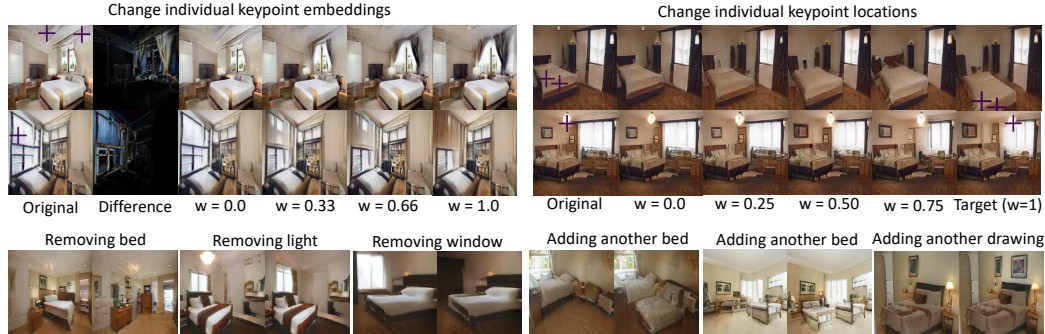

Figure 6: **Editing on Bedroom. Left first row:** interpolating the keypoint embedding on the curtain. **Left second row:** interpolating the keypoint embedding on the window. **Right first row:** changing the position of keypoint on the bed. **Right second row:** changing the position of the keypoints on the light. **Left third row:** removing parts from the bedroom. **Right third row:** adding parts.

which has an auto-encoder architecture and their reported FID is on reconstructed images instead of randomly generated appearances. In addition, the user study shows that on editing tasks, our method also exceeds the image editing quality of SEAN significantly (preferred by 94.78% of all responses).

## 4.5 GENERALIZATION TO OTHER DATASETS

**LSUN Bedroom and BBC Pose.** In Figure 6, we explore the editing ability of entire scenes on the LSUN bedroom dataset. No previous unsupervised part-based model has tried this difficult task before. We successfully interpolate the local appearance by changing the corresponding keypoint embeddings and translating the local key parts (window, bed) by moving the corresponding keypoints. We can also remove or add some parts by removing or adding peaks in the Gaussian Heatmaps. Since our learned representation is two-dimensional, it is not possible to rotate objects entirely. Figure 7 explores the editing of individuals. Although artifacts remain due to the detailed background and motion blur in the datasets, pose and appearance can be exchanged separately.

## 4.6 ABLATION TESTS

We test different variants of our architecture and demonstrate the effectiveness and necessity of our design. We test 1) removing the background; 2) removing the global style vector; 3) using additive global style vector instead of multiplicative ones; 4) using contrastive learned keypoint embeddings instead of multiplicative ones; 5) removing the keypoint embedding; 6) removing keypoints. We show the detection result on MAFL and FID score on FFHQ in Table 2. All contributions are crucial for the success and consistent for both detection and image quality tasks. The keypoint detection evaluation is explained in the subsequent Section 4.7. For simplicity, the FID is calculated at resolution $256 \times 256$, between 50k generated images and the original dataset; slightly different from previous experiments.

## 4.7 UNSUPERVISED KEYPOINTS DISCOVERY

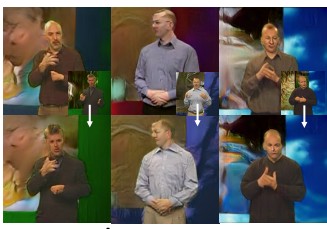 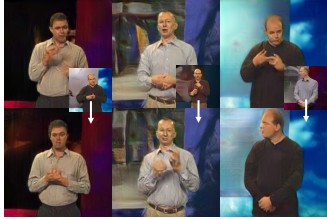 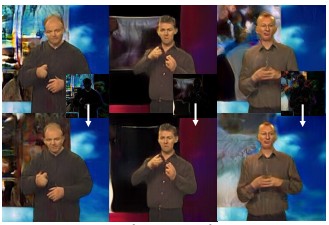

| Appearance | Pose | Background |

Figure 7: **Editing on BBC Pose.** The first row shows the source image and the second row the editing results. **First three columns:** the human appearance is swapped with the small target image. **Middle three columns:** changing the position to the one in the overlayed target. **Last three column:** changing the background (the bottom right corner shows the difference).

We demonstrated that our keypoints are semantically meaningful by editing the image via embeddings. In this part, we further demonstrate the consistency of our keypoints on the task of unsupervised keypoint detection as introduced by Thewlis et al. (2017). We test whether keypoints consistently align to the same image parts, without randomly moving on the image when other factors such as color or texture are changed. To this end, we train a standard ResNet detector (Xiao et al., 2018) supervised

| Method | $L_1$ error % ↓ | FID ↓ |
|---|---|---|
| full model | 5.85% | 23.50 |
| w/o background | 6.43% | 25.67 |
| w/o global style vector | 6.76% | 28.75 |
| adding global style vector | 5.29% | 42.02 |
| contrastive keypoint embedding | 7.53% | 28.47 |
| w/o keypoint embedding | 22.81% | 32.41 |
| w/o keypoint | - | 34.69 |

Table 2: **Quantitative ablation test** on keypoint localization ($L_1$) and image quality (FID). A lower number is better.

on 200,000 image-keypoint pairs generated on-the-fly by our LatentKeypointGAN. Note, the LatentKeypointGAN is trained on the same number of training images as the other detectors, hence, results are comparable. For evaluation, we follow Thewlis et al. (2017) and subsequent methods: As the order and semantics of unsupervised keypoints are undefined, a linear regressor from the predicted keypoints to the 5 ground truth keypoints is learned on the MAFL training set. All images are resized to $128 \times 128$ and the training and test set of the MAFL subset from CelebA are excluded when training the LatentKeypointGAN. The test error is the Euclidean distance to the ground truth on the MAFL test set, normalized by the inter-ocular distance in percent.

High accuracy can therefore only be obtained when estimated keypoints move consistently with the human-annotated ones. LatentKeypointGAN strikes a low error of 5.9%, which lies between the original 8.0% by Thewlis et al. (2017) and the most recent 2.8% by Dundar et al. (2020). We further tuned our architecture for precise localization by enforcing stronger translation equivariance of the keypoints (see Appendix B for details). This improved variant, LatentKeypointGAN-tuned, reaches 3.3%, which gets close to Dunbar et al. (2.8%). On the in-the-wild variant of CelebA (more diverse head orientation, position, and scale), which is significantly more difficult, we reach an error of 5.6%. Here, we outperform all existing methods reporting on this metric (8.7% by Jakab et al. (2018) and 11.4% by Lorenz et al. (2019)). We use the same hyperparameters as before, showing the strong robustness to new settings. Thereby, although not our main goal, LatentKeypointGAN also provides an alternative methodology for unsupervised keypoint/landmark detection that avoids paired training and TPS deformation, which are notoriously difficult to tune.

### 4.8 USER STUDY ON EDITING QUALITY

We conducted a user study to separately access image quality after editing, comparing against the unsupervised keypoint-based method (Zhang et al., 2018) on CelebA (Liu et al., 2015) and the mask-conditioned method (Zhu et al., 2020b) on CelebA-HQ (Liu et al., 2015). The study confirms the main results: While Zhang et al. strike a high keypoint accuracy, its image quality (ours preferred by 92.17%) and editing results (our disentanglement preferred by 67.83%) are unsatisfactory. While

Zhu et al. reach a better FID score before editing, after editing, ours is the most realistic (preferred by 94.78%). Moreover, our disentanglement comes close to the one of Zhu et al. (49.57% voted equal quality). Zhu et al. have better shape preservation because their masks provide silhouettes explicitly, but this likely also leads to the lower editing quality as masks and embeddings can become incompatible. For instance, the hair shape and strain direction need to be correlated. The full details are in the appendix.

## 5   LIMITATIONS AND FUTURE WORK

For portrait images, the hair can mix with the background encoding, such that it is considered part of the background. Still, the hair can be changed by selecting a background embedding with the desired hairstyle. Moreover, the disentanglement into locally encoded features can lead to asymmetric faces, such as a pair of glasses with differently styled sides. For BBC Pose, the keypoints are not well localized. They are consistent across images with the same pose, which permits pose transfer, but are too inconsistent for keypoint detection. Limitations could be overcome by linking keypoints hierarchically with a skeleton and by reasoning about masks on top of keypoints. While the face orientation in portrait images can be controlled, we found that orientation changes on the bedroom images are not reliable. The orientation is mostly baked into the appearance encoding. We believe that it will be necessary to learn a 3D representation and datasets with less-biased viewpoints.

## 6   CONCLUSION

We present a GAN-based framework that is internally conditioned on keypoints and their appearance encoding, providing an interpretable hidden space that enables intuitive editing. We provide the idea and implementation to avoid issues with TPS and related deformation models that are common in existing autoencoders. This LatentKeypointGAN also facilitates the generation of image-keypoint pairs, thereby providing a new methodology for unsupervised keypoint detection.

## 7   ETHICS STATEMENT

This research provides a new unsupervised disentanglement and editing method. By contrast to existing supervised ones, e.g., those requiring manually annotated masks, ours can be trained on any image collection. This enables training on very diverse sets as well as on personalized models for a particular population and can thereby counter biases in the annotated datasets.

Photo-realistic editing tools have the risk of abuse via *deep fakes*. A picture of a real person could be altered to express something not intended by the person. In that regard, our method has the advantage of only enabling the editing of generated images; it does not enable modifying real images; it only works in the synthetic to real direction. However, future work could extend it with an appearance encoder, which bears some risk.

Another risk is that the network could memorize appearances from the training set and therefore reproduce unwanted deformations of the pictured subjects. While Brock et al. (2018) and Karras et al. (2018) argue that GANs do not memorize training datasets, recently Feng et al. (2021) empirically proved that whether GANs memorize training datasets depends on the complexity of the training datasets. Therefore, our model has some risk of leaking such identity information if the training set is very small or the number of persons involved is limited, such as BBCPose. To mitigate the risk, we only use established and publicly available datasets, in particular those that collect pictures of celebrities or other public figures and also those not containing any person (bedroom dataset).

Our IRB approved the survey and we collected explicit and informed consent at the beginning of the study. Results are stored on secure servers. This research did not cause any harm to any subject.

On the technical side, we disclose all limitations in both the main paper and the supplementary. This research does not have conflicts with others. We credit the previous works in the Introduction (Section 1) and Related Work (Section 2).

## 8 REPRODUCIBILITY

We not only described our algorithm in Section 3 but also provide detailed information of our architecture, training strategy, and hyperparameters in the supplementary E. All the datasets we used are publicly available and the pre-processing is also described in Section 4. Furthermore, we provide the source code to facilitate future work in this direction of GAN-based generative image editing.

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

In this appendix, we present additional details on the neural network architectures, the progressive training, and hyperparameters. Furthermore, we show more qualitative results in the supplemental videos that are embedded and explained in the video/index.html.

## A    KEYPOINTS ILLUSTRATION

Figure 8 shows the keypoints generated on all datasets. The keypoints are semantically meaningful and very consistent across different instances.

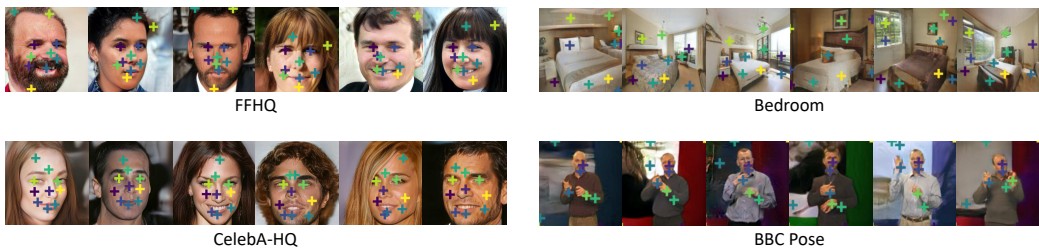

Figure 8: **Keypoints.** We show the keypoints on each dataset.

## B    TUNING LATENTKEYPOINTGAN FOR KEYPOINTS DETECTION

We desire an architecture that encodes the location of parts solely in the keypoint locations to improve keypoint localization and the subsequent learning of a detector. Even though the convolutional generator is translation invariant, additional care is necessary to prevent leakage of global position at the image boundary and from the starting tensor. All these tuning steps are explained below.

**Padding Margins.** As pointed out by Islam et al. (2020); Alsallakh et al. (2021); Xu et al. (2020a); Kayhan & Gemert (2020), convolutions with zero padding are very good at implicitly encoding absolute grid position at deeper layers. To prevent this, we follow Karras et al. (2021). By maintaining a fixed margin around the feature map and cropping after each upsampling, we effectively prevent the leaking of absolute position and the bias to the center because none of the generated pixels ever reaches a boundary condition. We use a 10-pixel margin. Note that such measures do not directly apply to autoencoders who are bound to the fixed resolution of the input.

**Positional Encoded Starting Tensor.** We remove the $4 \times 4 \times 512$ starting tensor because it can encode absolute grid position. We replace it with the positional encoding $\mathbf{M}$ of difference between keypoints $\mathbf{k}_1, ..., \mathbf{k}_K$ and the grid positions $\mathbf{p}$,

$$\mathbf{M}(\mathbf{p}) = [\sin(\pi * \text{Linear}([\mathbf{p} - \mathbf{k}_1, ..., \mathbf{p} - \mathbf{k}_K])), \cos(\pi * \text{Linear}([\mathbf{p} - \mathbf{k}_1, ..., \mathbf{p} - \mathbf{k}_K]))]. \quad (8)$$

The underlying idea is to only encode relative distances to keypoints but not to the image boundary.

**Larger Starting Tensor.** We found that starting the feature map from $32 \times 32$ instead of $4 \times 4$ improves keypoint localization accuracy.

**Background Handling.** The background should be consistent across the whole image, but complete occlusion is inaccurate to model with the sparse keypoints and their limited support in the intermediate feature maps. Hence, we introduce the explicit notion of a foreground mask that blends in the background. The mask is generated as one additional layer in the image generator. To generate the background, we use a separate network that is of equivalent architecture to LatentKeypointGAN-tuned. Note that in background generation we use AdaIN (Huang & Belongie, 2017) instead of SPADE because there is no spatial sensitive representation, such as keypoints. Foreground and background are then blended linearly based on the foreground mask. We use an $1 \times 1$ convolution layer to generate the final RGB images.

| Method | Aligned (K=10) | Wild (K=4) | Wild (K=8) |
|---|---|---|---|
| Thewlis et al. (2017) | 7.95% | - | 31.30% |
| Zhang et al. (2018) | 3.46% | - | 40.82% |
| Lorenz et al. (2019) | 3.24% | 15.49% | 11.41% |
| IMM (Jakab et al., 2018) | 3.19% | 19.42% | 8.74% |
| Dundar et al. (2020) | **2.76**% | - | - |
| LatentKeypointGan-tuned | 3.31% | **12.1**% | **5.63**% |
| - larger starting tensor | 3.50% | 14.22% | 8.51% |
| - separated background generation | 4.24% | 19.29% | 14.01% |
| - positional encoded starting tensor | 5.26% | 24.12% | 23.86% |
| - margin | 5.85% | 25.81% | 21.90% |

Table 3: **Landmark detection on CelebA (lower is better)**. The metric is the landmark regression (without bias) error in terms of mean $L_2$ distance normalized by inter-ocular distance. The bottom four rows shows our improvement step by step. We use the same number of keypoints as previous methods.

**Simplification** In addition, we remove the progressive training (Karras et al., 2018) and use the ResBlocks as defined by Park et al. (2019). This step is for simplification (quicker training results on medium-sized images) and does not provide a measurable change in keypoint accuracy.

As shown in Table 3, each step of improvement contributes significantly to the keypoint accuracy and consistent improvement on prior works on the two in-the-wild settings. Please see the main paper for the complete comparison. The FID of LatentKeypointGAN-tuned on CelebA of resolution $128 \times 128$ is 18.34.

## C    IMAGE EDITING QUALITY COMPARISON

Figure 9 validates the improved editing quality, showing comparative quality to conditional GAN (supervised by paired masks), and superior quality to unsupervised methods (greater detail in the hair and facial features). Note that for SEAN (Zhu et al., 2020b) we use the edited images (combine mask with different face appearance) instead of reconstructed images (combine mask with the corresponding face appearance) for fair comparison with other GAN methods. The qualitative improvement is further estimated in the subsequent user study.

The main paper reports FID scores for all those datasets where prior work does. To aid future comparisons we also report the remaining FID scores: 17.88 on FFHQ, 18.25 on CelebA, 30.53 on BBC Pose, and 18.89 on LSUN Bedroom. The FID is calculated by the generated 50k images and the resized original dataset.

We also qualitatively compare our method with a publicly available version[1] of Lorenz et al. (2019) on LSUN Bedroom, using the standard parameters that work well on other datasets. As shown in Figure 10, their model generates trivial keypoints and fails to reconstruct the bedroom images.

## D    EDITING QUALITY SURVEY

We designed comparisons in 4 different aspects to demonstrate our editing quality. We compare to two methods, one unsupervised keypoint-based method (Zhang et al., 2018) on CelebA (Liu et al., 2015), and one mask-conditioned method (Zhu et al., 2020b) on CelebA-HQ (Liu et al., 2015). For both methods, we follow their experiment design in their papers to make a fair comparison. For each question, we ask the participants to chose from 3 options: 1) ours is better; 2) theirs is better; 3) both methods have equal quality. The order of the pair to compare (ours in the first or the second) in the questions is randomly generated by the function `numpy.random.choice` of the package `Numpy` in `Python` with random seed 1. The results are illustrated in Table 4. The entire study is in anonymous form in the supplemental document at survey/index.html.

---

[1] `https://github.com/theRealSuperMario/unsupervised-disentangling/tree/reproducing_baselines`

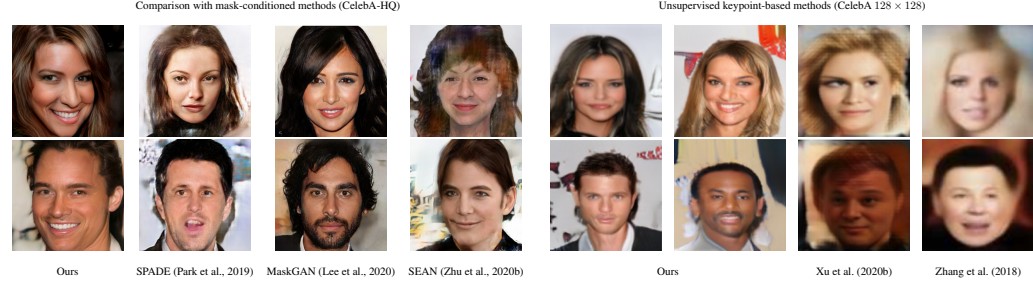

Figure 9: **Image editing quality comparison.** We compare the image editing quality with both, supervised (left) and unsupervised (right). LatentKeypointGAN improves on the methods in both classes.

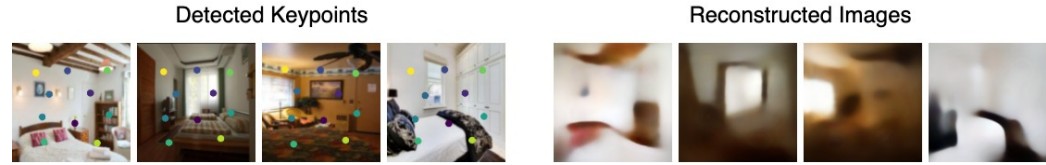

Figure 10: **Lorenz et al. (2019) on LSUN Bedroom.** (Left) Detected keypoints. The keypoints are static and do not have semantic meaning. (Right) Reconstructed images. The reconstruction completely fails.

**Study details.** We invited 23 participants answering 35 questions in total, with 5 for each of the categories above and an additional 5 comparing to randomly selected real images from the celebA training set. The generated images are selected as follows. We generate 32 images with LatentKeypointGAN. For the baselines we take the corresponding images from their paper (as implementations are not available). We then find our image (out of the 32) that best match the baseline images (in gender, pose, head size). If multiple images are similar, we use the first one.

**Editing image quality.** We edited the images by swapping the appearance embedding between different images. In each question, we show one image of ours and one image of theirs. We ask the participants to compare the resulting image quality.

**Part disentanglement.** To compare with Zhang et al. (2018), we moved part of the keypoints, as they did in their paper. To compare with Zhu et al. (2020b), we exchange the part embedding between different images, as they did in their paper. In each question, we show one pair of images of ours and one pair of images of theirs. We ask the participants to choose the one with better spatial disentanglement regardless of the image quality.

**Identity preservation while changing expression.** We compare identity preservation with Zhang et al. (2018). Following their paper, we change part (or all) of the keypoints to change the expression of the face. In each question, we show the two pairs of images. Each pair contains two images, one original image, and one edited image. We ask the participants to choose the pair that can better preserve the identity of the face regardless of the image quality, as quality is assessed separately.

**Shape preservation while changing appearance.** We compare the shape preservation with Zhu et al. (2020b). We edited the images by swapping the appearance embedding between different images. In each question, we show the two triplets of images. Each triplet contains three images, one shape source image and one appearance image, and one combined image. We ask the participants to choose the triplet where the combined image has the more similar shape as the shape source image regardless of the image quality.

**Interpretation - comparative.** This study confirms the findings of the main paper. Our method outperforms Zhang et al. in all metrics in Table 4. We also outperform SEAN in image editing

| Aspect | Method to compare | In favour of ours | In favour of others | Equal quality |
|---|---|---|---|---|
| Editing image quality | Zhang et al. (2018) | **92.17**% | 0.87% | 6.96% |
| Editing image quality | SEAN (Zhu et al., 2020b) | **94.78**% | 2.61% | 2.61% |
| Part disentanglement | Zhang et al. (2018) | **67.83**% | 5.22% | 26.95% |
| Part disentanglement | SEAN (Zhu et al., 2020b) | 28.69% | 21.74% | **49.57**% |
| Identity preservation | Zhang et al. (2018) | **55.65**% | 14.78% | 29.57% |
| Shape preservation | SEAN (Zhu et al., 2020b) | 33.91% | **46.96**% | 19.13% |

Table 4: **Survey results**. We compare 4 different aspects with other methods. The first one is the editing image quality. The second one is part disentanglement. The third one is identity preservation while changing expression. The last one is shape preservation while changing appearance.

quality. This confirms our claims of superior editing capability but may be surprising on the first glance since they attain a higher quality (better FID score) on unedited images. However, it can be explained with the limited editing capabilities of the mask-based approaches discussed next.

Participants give SEAN a higher shape preservation quality (47% in favour and 19% equal), which is expected since it conditions on pixel-accurate segmentation masks that explicitly encode the feature outline. However, the masks have the drawback that they dictate the part outline strictly, which leads to inconsistencies when exchanging appearance features across images. For instance, the strain direction of the hair and their outline must be correlated. This explains why our mask-free method attaining significantly higher image quality after editing operations (95% preferred ours). Hence, the preferred method depends on the use case. E.g., for the fine editing of outlines SEAN would be preferred while ours is better at combining appearances from different faces.

An additional strong outcome is that our unsupervised approach has equal disentanglement scores compared to SEAN; 50% judge them equal, with 29% giving preference to ours and only 22% giving preference to SEAN. Validating that LatentKeypointGAN enables localized editing.

**Interpretation - realism.** When comparing our GAN (without modifying keypoint location or appearance) to real images at resolution $128 \times 128$ of the training set, 42% rate them as equal. Surprisingly 33% even prefer ours over the ground truth. This preference may be because the ground truth images have artifacts in the background due to the forced alignment that are smoothed out in any of the generated ones. Overall, these scores validate that the generated images come close to real images, even though minor artifacts remain at high resolutions.

# E  EXPERIMENTS DETAILS

**SPADE** (Park et al., 2019) As shown in Figure 11, SPADE takes two inputs, feature map and style map, and use the style map to calculate the mean and standard deviation, which is used to denormalize the batch normalized feature map. Formally speaking, let $\mathbf{F}^i \in \mathbb{R}^{N \times C_i \times H_i \times W_i}$ be a $i$-th feature map in the network for a batch of $N$ samples, where $C_i$ is the number of channels. Here we slightly abuse the notation to denote $N$ batched style maps of size $(H_i, W_i)$ as $\mathbf{S}^i \in \mathbb{R}^{N \times (K+1)D_{\text{embed}} \times H_i \times W_i}$. The same equation as for BatchNorm (Ioffe & Szegedy, 2015) is used to normalize the feature map, but now the denormalization coefficients stem from the conditional map, which in our case is the processed style map. Specifically, the resulting value of the spatial adaptive normalization is

$$A^i_{n,c,y,x}(\mathbf{S}, \mathbf{F}) = \gamma^i_{c,y,x}(\mathbf{S}^i_n) \frac{\mathbf{F}^i_{n,c,y,x} - \mu^i_c}{\sigma^i_c} + \beta^i_{c,y,x}(\mathbf{S}^i_n), \qquad (9)$$

where $n \in \{1, ..., N\}$ is the index of the sample, $c \in \{1, ..., C\}$ is the index of channels of the feature map, and $(y, x)$ is the pixel index. The $\mu^i_c$ and $\sigma^i_c$ are the mean and standard deviation of channel $c$. The $\gamma^i_{c,y,x}(\mathbf{S}^i_n)$ and $\beta^i_{c,y,x}(\mathbf{S}^i_n)$ are the parameters to denormalize the feature map. They are obtained by applying two convolution layers on the style map $\mathbf{S}^i_n$.

**Learning rate and initialization.** We set the learning rate to 0.0001 and 0.0004 for the generator and discriminators, respectively. To let our model learn the coordinates reliably, we first set the learning rate of the MLP, which generates keypoint coordinates to 0.05x the generator learning rate,

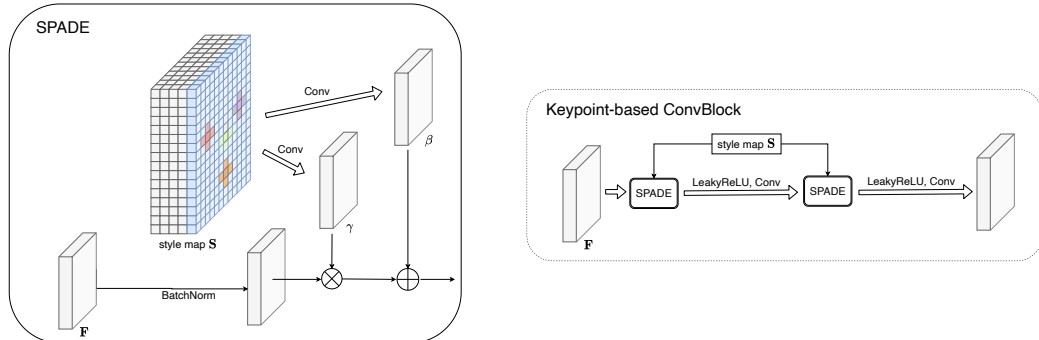

Figure 11: (Left) **Spatially Adaptive Normalization.** SPADE takes two inputs, feature map and style map. It first uses batch normalization to normalize the feature map. Then it uses the style map to calculate the new mean map and new standard deviation map for the feature map. (Right) **Keypoint-based ConvBlock.**

i.e. 0.00005. We use Kaiming initialization (He et al., 2015) for our network. We initialize the weights of the last layer of the MLP that generates the keypoint coordinates to 0.05x the Kaiming initialization.

**Progressive Growing Training**    we adopt progressive growing training (Karras et al., 2018) to allow larger batch size, which helps both on keypoint localization and local appearance learning. This is likely due to the BatchNorm that is essential in SPADE. We also tried to replace the BatchNorm in SPADE with LayerNorm (Ba et al., 2016) and PixelNorm (Karras et al., 2019), but both of them cause mode collapse. We use a scheduled learning rate for the Keypoint Generator $\mathcal{K}$. As illustrated in Figure 12, at each resolution stage, the training is divided into adapting period and non-adapting period. We set the learning rate of $\mathcal{K}$ to zero in the adapting period and back to normal in the non-adapting period. In the adapting period, the training follows PGGAN (Karras et al., 2018) where the feature map is a linear combination of the larger resolution RGB image and current resolution RGB image. The coefficient $\alpha$ gradually increases from 0 to 1. At the end of the adapting period, the network is fully adapted to generate higher resolution images. In the non-adapting period, the network generates high-resolution images without the linear combination. Following StyleGAN (Karras et al., 2019), we start from a $4 \times 4 \times 512$ learned constant matrix, which is optimized during training and fixed during testing. We use the keypoint-based ConvBlock 2 and bilinear upsampling to obtain feature maps with increasing resolutions. Unlike PGGAN (Karras et al., 2018) and Style-GAN (Karras et al., 2019), who generating RGB images from feature maps of all resolutions (from $4 \times 4$ to $1024 \times 1024$), we start generating RGB images from the feature maps of at least $64 \times 64$ resolution. This is possible with the keypint generator and its spatially localized embeddings taking over the role of low feature maps. It helps to locate the keypoints more accurately.

**Generator.**    We illustrate the LatentKeypointGAN generator in Figure 12. The output image is linearly combined by the output of *toRGB* block, where the weights depend on the training stage.

**Discriminator.**    We illustrate the discriminator in Figure 12. For each resolution, we use two convolutions followed by Leaky ReLU (Maas et al., 2013). The first convolution has a kernel size $4 \times 4$ and stride 2 to downsample the feature map to 0.5x. The second convolutions have a kernel size $3 \times 3$ and stride 1 to extract features.

**Hyperparameter Setting**    For all experiments in image generation, we use leaky ReLU (Maas et al., 2013) with a slope 0.2 for negative values as our activation function. We use ADAM optimizer (Kingma & Ba, 2015) with $\beta_1 = 0.5$ and $\beta_2 = 0.9$. We set the learning rate to 0.0001 and 0.0004 for generator and discriminators, respectively (Heusel et al., 2017). We start from generating $64 \times 64$ images in the progressive training. The batch size for $64^2, 128^2, 256^2, 512^2$ images are $128, 64, 32, 8$, respectively. We set $\lambda_{gp} = 10$ and $\lambda_{bg} = 100$. We set $D_{noise} = 256$ and $D_{embed} = 128$ for all experiments unless otherwise stated (in ablation tests).

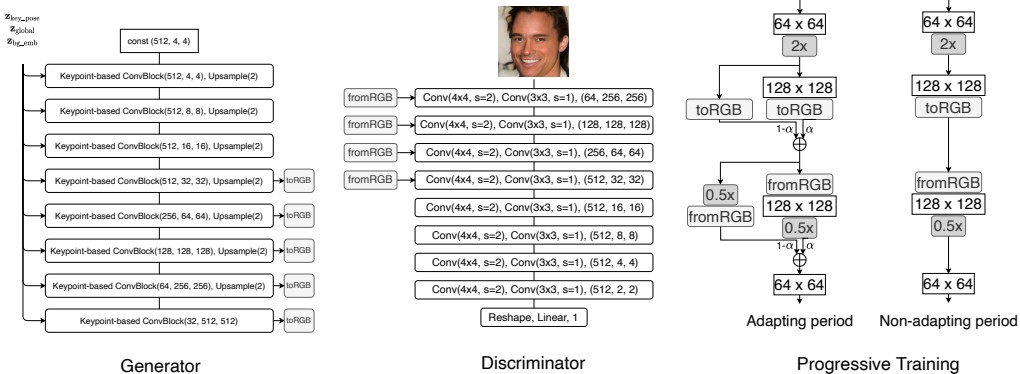

Figure 12: **Detailed architecture.** (Left) **LatentKeypointGAN generator.** The numbers in the parenthesis is the output dimension of the Keypoint-based ConvBlock. For example, (512, 4, 4) means the output feature map has a resolution of $4 \times 4$ and the channel size is 512. The toRGB blocks are $1 \times 1$ convolutions to generate the RGB images with the same resolution as corresponding feature maps. (Middle) **LatentKeypointGAN discriminator.** The number in the last parenthesis is the output dimension. For example, (512, 4, 4) means the output feature map has a resolution of $4 \times 4$ and the channel size is 512. At each resolution, we apply two convolutions, one with stride 2 to downsample feature maps and one with stride 1 to extract features. Leaky ReLU (Maas et al., 2013) is used after all convolutions except the linear layer in the last.. (Right) **Progressive Training..** The adapting period is the same as PGGAN (Karras et al., 2018) and StyleGAN (Karras et al., 2019). In the non-adapting period, we do not use the linear combination.

| Dataset | background module and loss | $\tau$ |
|---------|---------------------------|--------|
| CelebA-HQ | yes | 0.01 |
| FFHQ | yes | 0.01 |
| Bedroom | no | 0.01 |
| BBC Pose | yes | 0.025 |

Table 5: **Setting for different datasets**. For the Bedroom dataset, we do not use the background module and loss. For the BBC Pose dataset, we use $\tau = 0.025$.

We lists the different $\tau$s and different background setting for all experiments in Table 5. In CelebA-HQ and FFHQ, the foreground is naturally disentangled from the background. The face can be freely moved on the image. However, in the Bedroom dataset, all objects and their parts are strongly correlated. For example, the bed cannot be moved to the ceiling, and the window cannot be moved to the floor. Therefore, we treat every object in the bedroom as a key part, even the floor, but the possible motion is restricted to plausible locations (see the supplementary video). A separate background embedding does not make sense. Therefore, we set the background ($\mathbf{H}^{bg} = 0$) and the background loss $\lambda_{\text{bg}} = 0$ for the experiments on the Bedroom dataset.

## F    ABLATION TESTS

### F.1    ABLATION TEST ON THE NEURAL NETWORK ARCHITECTURE

We show the detection result on MAFL and FID score on FFHQ in Table 2 for different architectures we tested below. The detection follows Section 4.7. The FID is calculated at resolution $256 \times 256$, by generated 50k images and the resized original dataset. Note that it is different from the main paper for simplicity since the goal of this part is ablation tests instead of comparing with others.

**Removing background embedding.**    We remove the background embeddings from our architecture ($z_{\text{bg\_emb}}$ and $\mathbf{w}_{\text{bg}}$ in Figure 2). In this case, the keypoint embedding controls the whole appear-

ance of the image. In addition, as shown in Figure 14, the keypoints are not exactly located at the key parts, though they are still consistent among different views.

**Removing global style vector.** We remove the global style vector $\mathbf{w}_{\text{global}}$. Therefore, all the keypoint embeddings are constant. Only keypoint location and background embedding are different among the images. In this case, the keypoint embedding works equivalent to one-hot encoding, and cannot fully capture the variations on the key parts. Therefore, it leads to inaccurate keypoints, as shown in Figure 14. Furthermore, we observed that without $\mathbf{w}_{\text{global}}$, the network hides the appearance information in the keypoint location.

**Changing keypoint embedding generation.** We change the keypoint embedding generation in two ways. The first way is generating constant embedding $\mathbf{w}_{\text{const}}^j$ and global style vector $\mathbf{w}_{\text{global}}$ just as before and then add them elementwisely instead of multiplying them. Formally speaking, for each keypoint $j$, its corresponding embedding is

$$\mathbf{w}^j = \mathbf{w}_{\text{global}} \oplus \mathbf{w}_{\text{const}}^j, \tag{10}$$

where $\oplus$ means elementwise addition. This gives slightly higher detection accuracy but lower image quality. We observe that in this case, the background controls the foreground appearance. However, different from Removing global style vector F.1, the appearance information is not hidden in keypoint locations. We believe this is because that $\mathbf{w}_{\text{global}}$ works as noise to avoid the network from hiding foreground appearance information in keypoint location. As a result of good disentanglement of appearance and keypoint location, the keypoint detection accuracy slightly increases. However, again, in this setting, the keypoint embedding cannot fully capture the variations of the key parts. Therefore, the background takes the control of appearance.

The second way is to generate $[\mathbf{w}^j]_{j=1}^K$ together from $\mathbf{z}_{\text{kp\_app}}$ using MLP. In this case, there is no constant embeddings or global style vector. To force the embedding of the same keypoint to be similar, and the embedding for different keypoints to be different, we use Supervised Contrastive Losses (Khosla et al., 2020),

$$\mathcal{L}_{\text{contrastive}}(\mathcal{G}) = -\sum_{j \in J} \frac{1}{|K(j)|} \sum_{k \in K(j)} \log \frac{\exp(\mathbf{w}^j \cdot \mathbf{w}^k / T)}{\sum_{a \in A(j)} \exp(\mathbf{w}^j \cdot \mathbf{w}^a / T)}, \tag{11}$$

where

$$A(j) = \{i : \mathbf{w}^i, \mathbf{w}^j \text{ are in the same batch}\}$$
$$K(j) = \{i : \mathbf{w}^i, \mathbf{w}^j \text{ belong to the same keypoint in the same batch}\}$$
$$J = \{\text{indices of all keypoint embeddings in the same batch}\}$$

As shown in Figure 14, the keypoints are neither on the key parts nor consistent. We further visualize the embeddings with T-SNE and PCA in Figure 13. Although the contrastive learned embedding has comparable T-SNE with our multiplicative design, the PCA shows that our multiplicative embedding is linearly separable while contrastive learned embedding is not. Hence, we demonstrate that our original design of elementwise production is simple and effective.

**Removing keypoint embedding.** We remove the keypoint embedding $\mathbf{w}^j$ entirely. In this case, we only have background embedding $\mathbf{w}_{\text{bg}}$ and the keypoint location. Thus, instead of generating the style map $\mathbf{S}$, we directly concatenate the keypoint heatmaps $\{\mathbf{H}^k\}_{k=1}^K$ and the broadcasted background style map to generate the style map without keypoint embedding. As shown in Figure 14, the keypoints are not meaningful or consistent. The keypoint location hides the appearance and is entangled with the background severely.

**Removing keypoints.** If we remove the keypoints, then SPADE (Park et al., 2019) degenerates to AdaIN (Huang & Belongie, 2017). Instead of using a style map $\mathbf{S}$ (2D), we now use a style vector (1D), which is the background embedding. In this case, we do not have the power of local controllability on the generated images.

## F.2 Ablation Tests on the Hyperparameters

**Ablation test on the dimension of embeddings.** Different numbers of embedding dimensions make the expression power vary. As shown in Table 6, larger $D_{\text{embed}}$ leads to larger error on MAFL

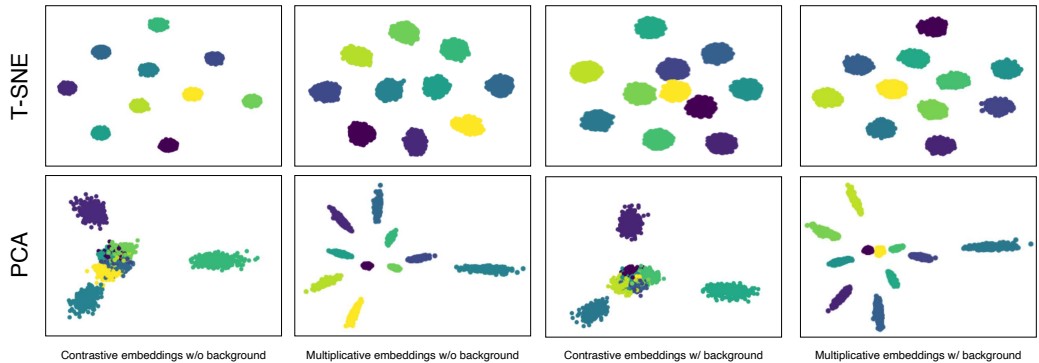

Figure 13: **Ablation study on multiplicative embedding**. We show the T-SNE and PCA visualization of embeddings learned on FFHQ. The first two column shows keypoint embeddings and the last two column shows keypoint embeddings and background embedding.

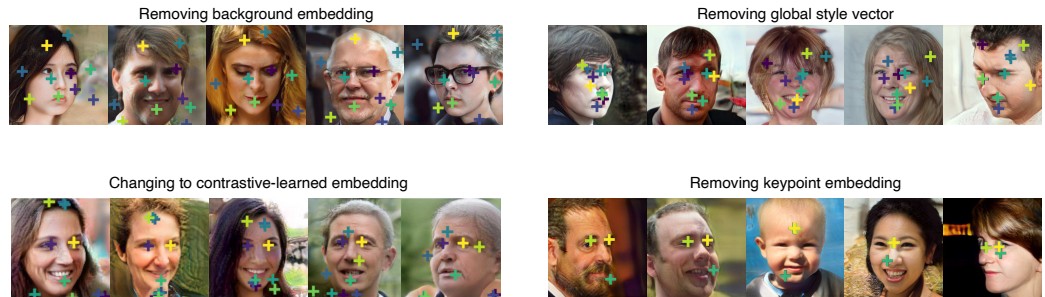

Figure 14: **Ablation study on architecture**. We show the keypoints for different architectures.

but lower (better) FID. We use $D_{embed} = 128$ in our main paper because the increase in error is small but the decrease of FID is significant.

**Ablation Test on $\tau$**    A too-small value for $\tau$ does not influence the image and will cause artifacts as shown in Figure 15. A too-large value for $\tau$ will disable the background embedding and control the background.

**Ablation test on the number of keypoints.**    By selecting different numbers of keypoints, we can achieve different levels of control. In the second row of Figure 15, we use 6 keypoints rather than the default 10. Thereby, keypoints have a smaller area of effect. We observe that the background encoding then takes a larger role and contains the encoding of hair and beard, while the keypoints focus only on the main facial features (nose, mouth, and eyes).

**Ablation test on the combination of number of keypoints and $\tau$.**    The impact of keypoints depends on the combination of number of keypoints and $\tau$. We test the pairwise combination between $K = 1, 6, 8, 12, 16, 32$ and $\tau = 0.002, 0.005, 0.01, 0.02$. The FID is listed in Table 7 and the detec-

| Dimension of embeddings $D_{embed}$ | Keypoint detection error on MAFL ↓ | FID score on FFHQ ↓ |
|---|---|---|
| 16 | 4.61% | 28.85 |
| 32 | 4.92% | 26.14 |
| 64 | 5.66% | 27.64 |
| 128 | 5.85% | 23.50 |

Table 6: **Quantitative ablation test on dimension of embeddings.** For both metrics, the lower means better.

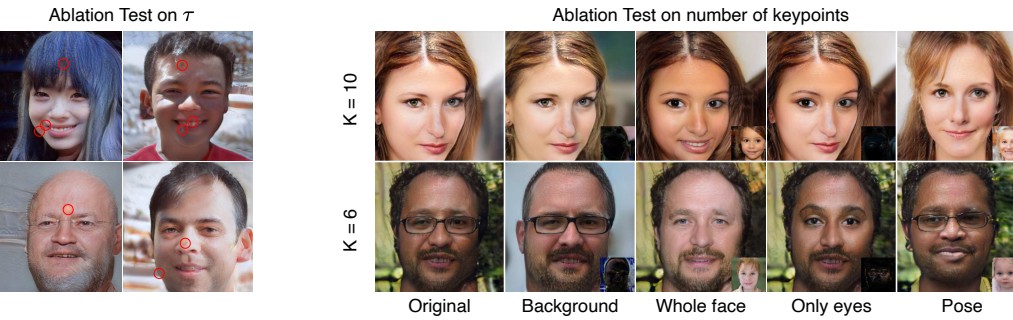

Figure 15: **Ablation study on hyperparameters**. (Left) Face generation with on FFHQ with $\tau = 0.002$. We use the red circle to mark the artifacts in the images. (Right) Face generation on FFHQ with number of keypoints 10 (top) and 6 (bottom). More keypoints lead to a stronger influence of the keypoint embedding. However, the 6-keypoint version still provides control, e.g., glasses, nose type, and pose. From left to right: original image, replaced background (difference map overlaid), replaced keypoint embeddings (target image overlaid), exchanged eye embeddings, and keypoint position exchanged.

|            | $K = 1$ | $K = 6$ | $K = 8$ | $K = 12$ | $K = 16$ | $K = 32$ |
|------------|---------|---------|---------|----------|----------|----------|
| $\tau = 0.002$ | 19.35 | 18.94 | 17.77 | 17.69 | 20.44 | 19.29 |
| $\tau = 0.005$ | 18.39 | 18.28 | 18.42 | 18.38 | 20.31 | 18.72 |
| $\tau = 0.01$  | 19.49 | 19.60 | 20.31 | 18.14 | 19.25 | 17.91 |
| $\tau = 0.02$  | 19.11 | 18.80 | 20.28 | 19.34 | 19.58 | 18.17 |

Table 7: **FID scores of ablation tests on number of keypoints $K$ and keypoint size $\tau$ on CelebA of resolution $128 \times 128$.** The lower means better. Neither $K$ or $\tau$ significantly influence the image quality. Interestingly, the small artifacts when $\tau = 0.002$ in Figure 15 does not neither significantly influence the image quality.

tion error is listed in Table 8. The image quality does not change much for different combinations. We illustrate samples of keypoints of each combination in Figure 16 and editing results in Figure 17 and Figure 18. If both the number of keypoints and $\tau$ are small, e.g., $K = 1$, and the $\tau = 0.002$, then the background controls both foreground appearance and pose, and the keypoints are trivial. If both, the number of keypoints and $\tau$, are large, e.g., $K = 32$, and $\tau = 0.02$, then the keypoint appearance controls background and pose. While the model degenerates in extreme cases, we found the model to be robust for a wide range of values, i.e., $K = 8, 12, 16$ and $\tau = 0.005, 0.01$. We summarize all the cases in Table 9.

## F.3 Ablation Tests on the Training Strategy

**Ablation Test on GAN Loss** If we replace equation 3, 4 and 5 with the spectral norm (Miyato et al., 2018) and hinge loss (Miyato et al., 2018; Park et al., 2019) used in the original SPADE architecture, we get mostly static, meaningless latent keypoint coordinates. The object part location information is entangled with the key part appearance. The comparison is shown in Figure 21.

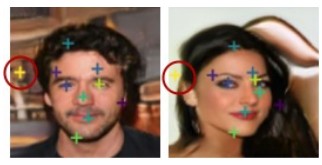

Figure 20: **Ablation Test on Background loss.**

**Ablation Test on Background Loss** If we remove the background loss in Eq. 6, most keypoints are still at reasonable positions while some move to the background. As shown in Figure 20, the yellow keypoint is on the background while all the others are still on the foreground.

**Removing Keypoint Scheduler.** If we move the keypoint scheduler, i.e., updating keypoint generator during resolution adaption, the keypoint locations diverge and the appearance collapses, as shown in Figure 21.

|  | $K = 1$ | $K = 6$ | $K = 8$ | $K = 12$ | $K = 16$ | $K = 32$ |
|---|---|---|---|---|---|---|
| $\tau = 0.002$ | 11.59% | 7.86% | 6.78% | 7.09% | 6.00% | 5.35% |
| $\tau = 0.005$ | 8.65% | 7.28% | 6.39% | 5.24% | 5.13% | 4.11% |
| $\tau = 0.01$ | 8.43% | 7.91% | 7.97% | 6.06% | 7.37% | 8.84% |
| $\tau = 0.02$ | 8.71% | 6.26% | 7.13% | 5.16% | 6.80% | 8.26% |

Table 8: **Normalized Error of ablation tests on number of keypoints $K$ and keypoint size $\tau$ on CelebA of resolution** $128 \times 128$**.** For $\tau = 0.002, 0.005$, the error decreases as $K$ increases while for $\tau = 0.01, 0.02$, the error first decreases and then increases. If both of them are large, e.g., $K > 16, \tau > 0.01$, the appearance is entangled with the keypoints which results in a larger error.

|  | $K = 1$ | $K = 6$ | $K = 8$ | $K = 12$ | $K = 16$ | $K = 32$ |
|---|---|---|---|---|---|---|
| $\tau = 0.002$ | T | T | T | T | T | T |
| $\tau = 0.005$ | T | ✓✓ | ✓✓ | ✓✓ | ✓✓ | ✓✓ |
| $\tau = 0.01$ | T | ✓ | ✓ | ✓✓ | ✓✓ | E |
| $\tau = 0.02$ | ✓ | ✓ | ✓ | ✓ | E | E |

Table 9: **Keypoint controllability**. T denotes trivial keypoint, i.e., the background controls the entire image. E means entangled pose, appearance and background. ✓✓ means disentangled control and ✓ means inferior disentanglement, where one of the pairs {(pose, appearance), (pose, background), (appearance, background) is entangled.}. For a small keypoint size of $\tau = 0.0002$ the model always gives trivial keypoints. With a large number of keypoints and a large keypoint size, i.e., $K > 16$ and $\tau > 0.01$, our model gives entangled representations. Our model is robust in the range of $K \in [8, 16]$ and $\tau \in [0.005, 0.01]$.

## G  FAILURE CASES AND LIMITATIONS

As described in the main text, our model sometimes generates asymmetric faces as shown in the first two images in Figure 22. In addition, the hair sometimes is entangled with the background, especially long hair, as shown in the right two images in Figure 22.

From the ablation tests in training strategy, we can see that this method can heavily depend on the state-of-the-art GAN training loss function and image-to-image translation architectures. In fact, we observed some image quality degeneration as the training goes on in the highest resolution ($512 \times 512$). Therefore, we apply early stopping in the highest resolution. We expect that researchers will push GAN and spatially adaptive image-to-image translation even further. Then people can plug in our keypoint generator to receive higher image quality and more accurate keypoints. In this paper, we only provide the idea and focus on unsupervised local image editing without using any loss on pairs of images.

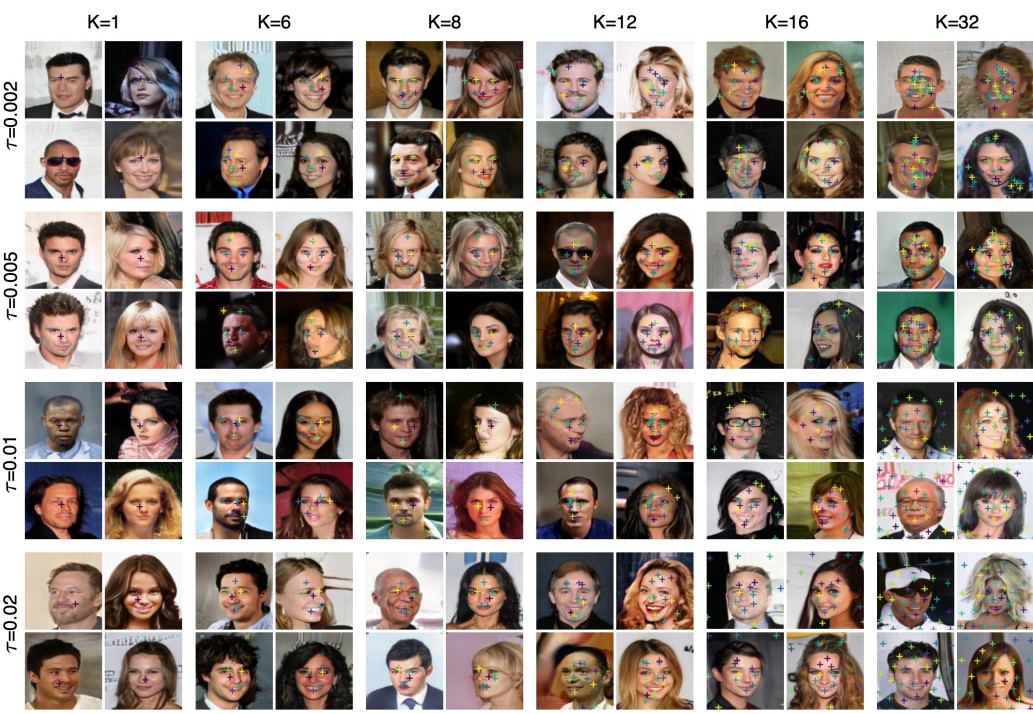

Figure 16: **Visualization for different combinations of number of keypoints and keypoint size** $\tau$. If both, the number of keypoints and the keypoint size $\tau$, are small (top left), the keypoint is trivial. If both of them are large (bottom right), the keypoints distribute uniformly over the images instead of focusing on parts.

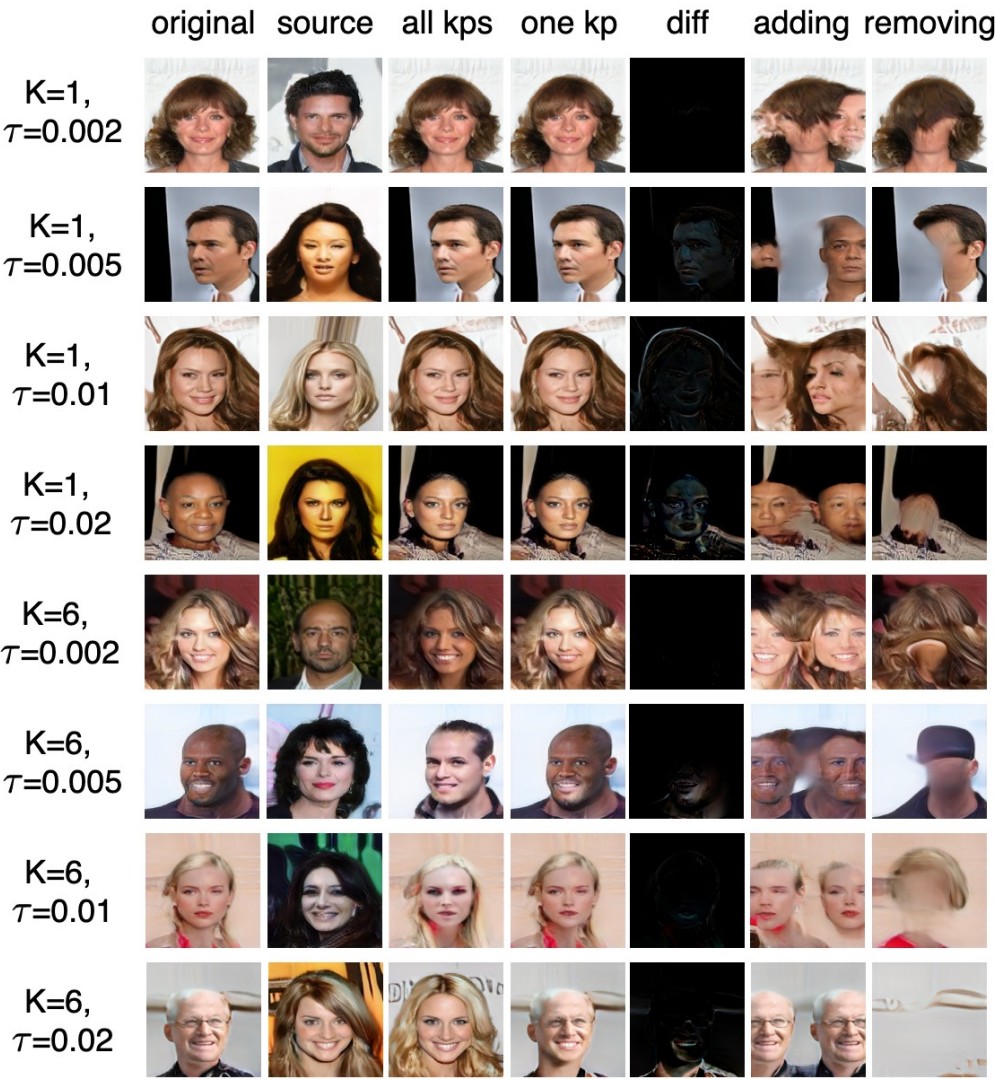

Figure 17: **Editing on different combinations of number of keypoints $K$ and keypoint size $\tau$.** K=1, 6. **Column 1**: original image; **column 2**: part appearance source image used to swap appearance; **column 3**: the combined image with shape from the original images and the appearance from the part appearance source image; **column4**: we randomly swap a single keypoint close to the mouth; **column 5**: resulting difference map when changing the keypoint in the 4th column; **column 6**: move the face to the left and add another set of keypoints on the right; **column 7**: removing all keypoints. If $\tau = 0.0002$, the keypoints are trivial, and cannot be used to change appearance. When $K = 1$, the keypoint also only have limited control even if $\tau = 0.02$. The combination of $K = 6, \tau = 0.005$ gives good spatial disentanglement.

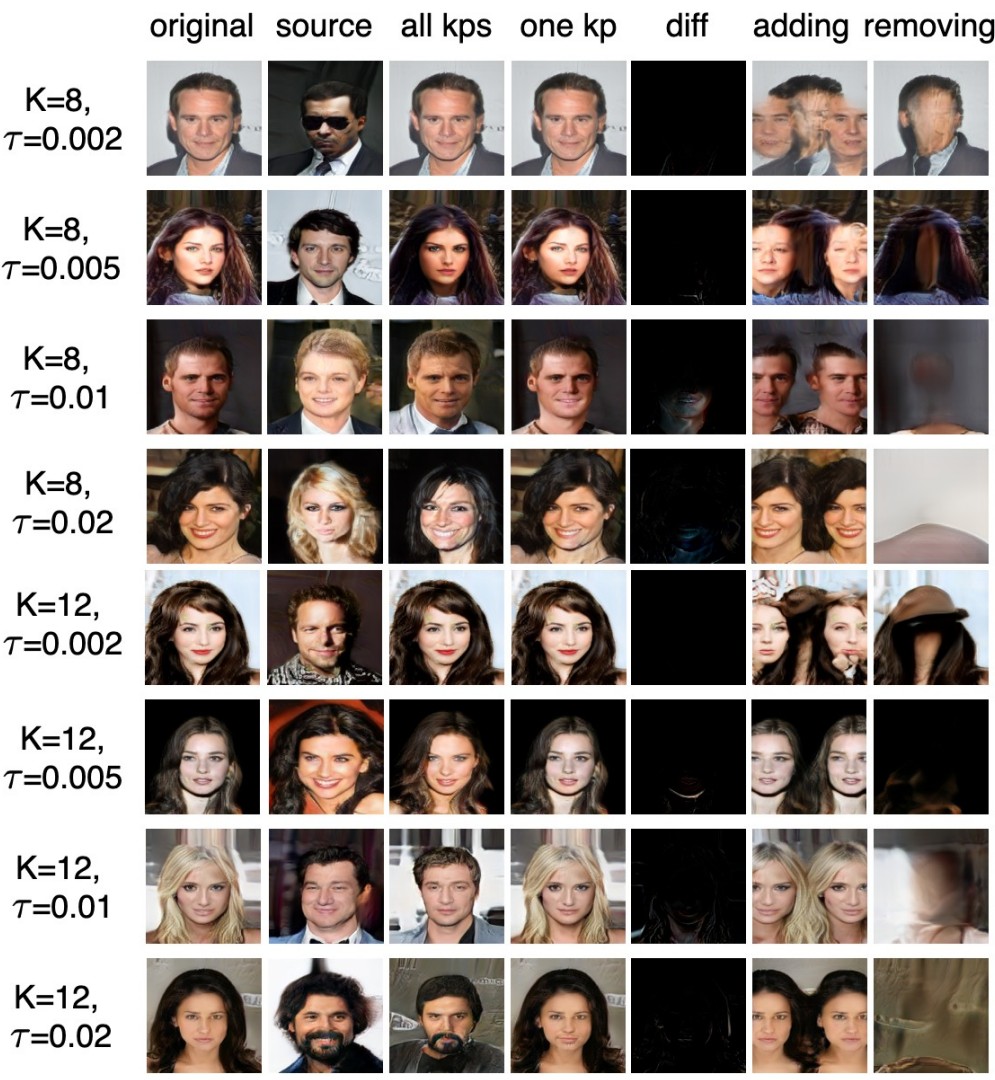

Figure 18: **Editing on different combinations of number of keypoints $K$ and keypoint size $\tau$.** K=8,12. For a small $\tau = 0.0002$, the keypoint is trivial. When $\tau$ is large the background is entangled ($K = 8, \tau = 0.02$) in some cases. We found the combinations of ($K = 12, \tau = 0.0005$) and ($K = 12, \tau = 0.01$) both give the best editing controllability.

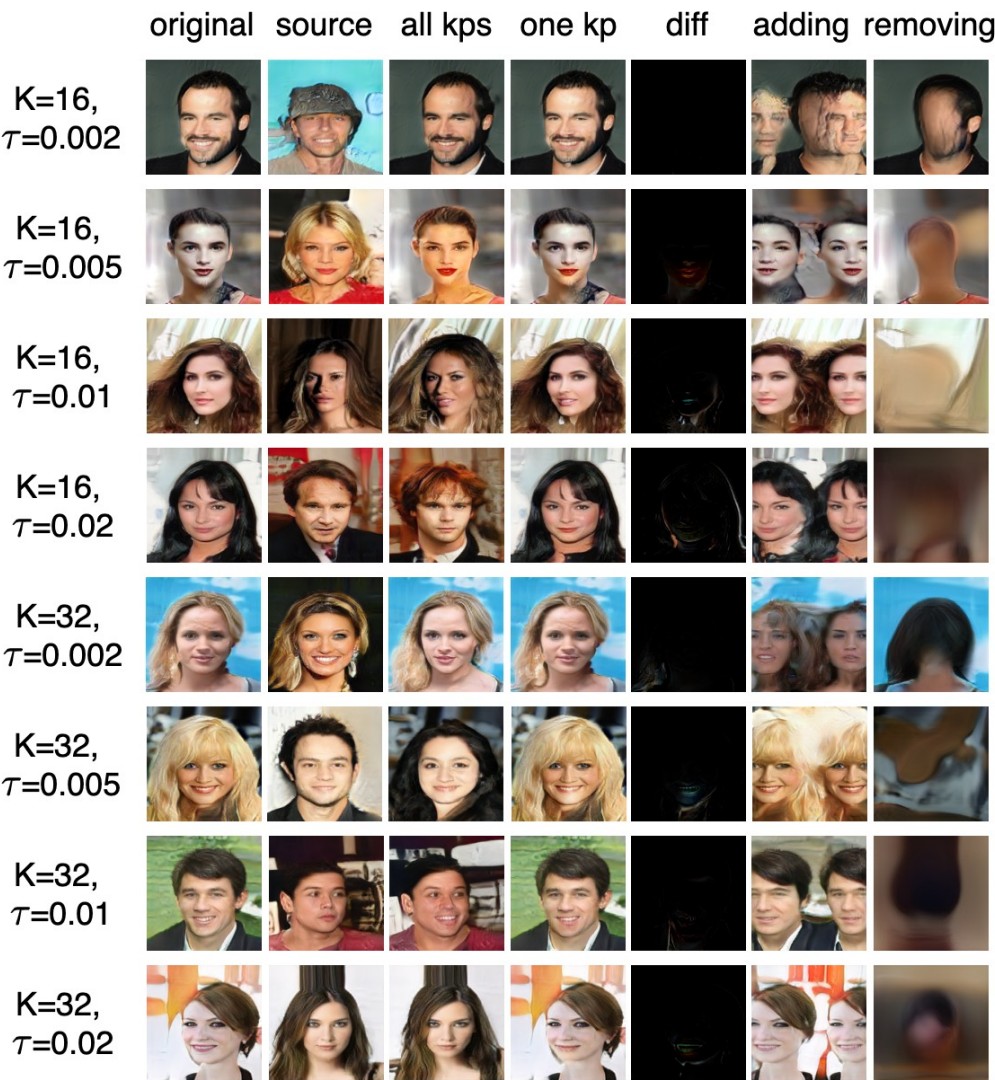

Figure 19: **Editing on different combination of number of keypoints $K$ and keypoint size $\tau$.** K=16,32. Extreme small $\tau$ ($\tau = 0.0002$) constantly gives trivial keypoints even if $K$ is large ($K = 32$). When both, $K$ and $\tau$, are large ($K = 32, \tau > 0.01$), the keypoint embeddings control the background and the pose. We found the combinations of ($K = 16, \tau = 0.0005$) and ($K = 32, \tau = 0.005$) both give the best editing controllability.

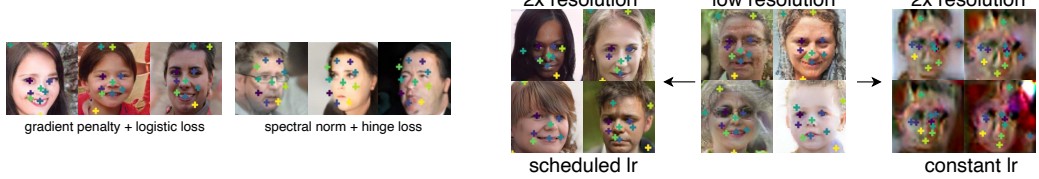

Figure 21: (Left) **GAN Loss Importance.** Without gradient penality + logistic loss, as in SPADE, keypoint coordinates remain static. (Right) **Scheduling the keypoint generator learning rate.** Reducing the learning rate after each progressive up-scaling step prevents mode collapse and enables high-resolution training.

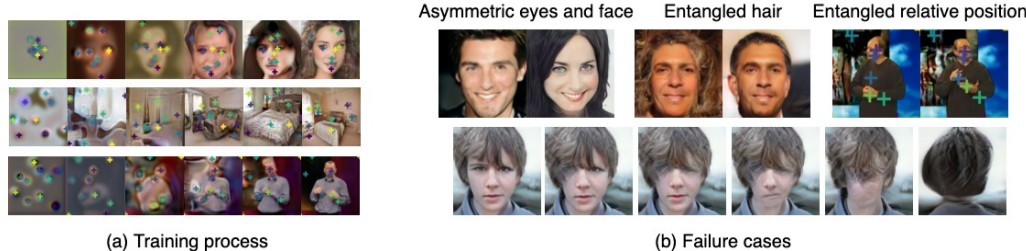

(a) Training process          (b) Failure cases

Figure 22: (a) **Training process.** We visualize the image generated during the training. (b) **Failure cases.** The left top two images show asymmetric faces: different eye colors for the man and different blusher for the woman. The middle top two images show the entanglement of hair and background. The right top two images show that the pose of head is hidden in the relative positions of other keypoints than the keypoints on the head. We visualize the process of removing parts at the bottom. We sequentially remove the left eye, right eye, mouth, nose, and the entire face. Due to the entanglement of hair and background, the hair remains even if we remove the whole face.

