# OpenReview forum: "LatentKeypointGAN: Controlling GANs via Latent Keypoints"
_ICLR.cc/2022/Conference — ICLR 2022 Submitted_

### Official Review · Reviewer_X57n · 2021-10-22

**Correctness:** 4
**Technical Novelty And Significance:** 2
**Empirical Novelty And Significance:** 1
**Recommendation:** 5
**Confidence:** 4

**Main Review:**

Strength:
1. The paper is well-written and easy to follow.
2. The evaluation is thorough and convincing.

Weakness:
1. The proposed method is not quite novel. Although the paper emphasized key points, the incorporation of $w_{const}^j$ and the spatial embedding layer (section 3.2) that broadcasts $w^j$ according to a simple positional heatmap implicitly change the key points to a special kind of "segmentation mask". In other words, the core method (section 3.1 and section 3.2) is essentially a conversion from key points to a special kind of "segmentation mask" that is Gaussian-like and does not have sharp boundaries. This also explains why it can be easily incorporated into SPADE.
2. As an unsupervised method, the performance of the proposed method depends heavily on two hyperparameters: the choice of the number of key points and their influence range $\tau$. These are not highlighted in the main paper but are slightly discussed in F.2 of the appendix. I believe the choice of these two hyperparameters interferes with each other and can be difficult to tune, which can make the method less useful. I would suggest adding an in-depth analysis of how these two hyperparameters influence the performance.

**Summary Of The Paper:**

This paper proposed a new way to disentangle and control the GAN synthesis process via key points positions, key points appearance and background appearance. Since the key points are learnt in an unsupervised way, the proposed method avoids the labour-intensive labelling process. Extensive experiments verify the effectiveness of the proposed method.

**Summary Of The Review:**

As abovementioned, I think the paper is well-written and polished to a high level. However, its contribution is a bit thin and it lacks an in-depth analysis of the two deterministic hyperparameters: the number of key points and their influence range $\tau$ (the discussion in appendix F.2 is not sufficient). Thus, I feel slightly negative on the submission.

---

> ### Author Response · Authors · 2021-11-23
> **Rebuttal X57n**
>
> Thank you for your time and substantial question. Please do not miss our general notes including hundreds of new qualitative results that were requested by other reviewers and validate those fewer ones in the main paper.
>
> __What is the contribution as Gaussian heatmaps are just smoothed masks?__
>
> A simple replacement of segmentation masks in SPADE with Gaussians was our starting point but insufficient to attain the goals. The difficulty lies in generating embeddings that can be used in a SPADE-like network, are disentangled, and enable positional editing control; all this in the unsupervised setting. Please check our ablation tests on keypoint embedding generation types in Appendix F1, Table 2, and Figure 13 to see the important of our contributions:
>  * If we remove the global style vector, the generated keypoints will be inaccurate and the network will hide appearance information in the keypoint location.
>  * If we use contrastive learning for embeddings, the generated keypoints are not consistent and the embeddings are not separable in the latent space.
>  * If we use additive embeddings, the image quality significantly decreases.
>  * If we simply use Gaussian heatmaps without embeddings, the keypoints are not meaningful or consistent. The keypoint location hides the appearance and the foreground is entangled with the background severely.
>
> Creating a suitable architecture is only the first step and we are glad that we can build upon the advances made by SPADE and other supervised approaches. Training in the unsupervised setting is a non-trivial extension. We provide rigorous empirical experiments to validate our hyperparameter choices and their robustness. For more details, please refer to Appendix F1.

---

> > ### Comment · Reviewer_X57n · 2021-11-29
> > **Thanks for the response**
> >
> > Thanks for the response and additional experiments. However, the rebuttal did not change my view of the paper.
> >
> > I understand that the proposed method aims to find the controls in an unsupervised way, which is not a trivial problem. However, the proposed method is a bit "trivial" and less useful as the controls and their semantic meanings are heavily dependent on the choice of the two hyper-parameters of unsupervised learning. Thus, I still feel slightly negative about the paper but I will not be bothered if it gets accepted.

---

### Official Review · Reviewer_Jmzu · 2021-10-24

**Correctness:** 2
**Technical Novelty And Significance:** 3
**Empirical Novelty And Significance:** 2
**Recommendation:** 6
**Confidence:** 4

**Details Of Ethics Concerns:**

Allowing to edit images on keypoint level takes a step towards more realistic DeepFakes. The authors address this concern to some extent in their statement but this might not be sufficient (see "Ethics statement" in main review).

**Main Review:**

### Strengths:
+ The paper is well-written and easy to follow. The main claim of allowing to edit images via keypoints is well-supported in the experimental section and in the videos in the supplementary material.
+ The paper includes many ablations to validate the architectural choices made. Overall, the proposed method appears technically sound.
+ The paper shows impressive results on LSUN beds which is a difficult dataset that none of the existing unsupervised approaches has tackled so far.
Quantitative evaluation on celebA-HQ suggests that the method produces similar image fidelity to methods that require more supervision.

### Weaknesses:
- There are not enough qualitative and quantitative comparisons to the baseline approaches. The method claims that existing works struggle with the disentanglement of pose and appearance, but the experimental section does not validate that their method outperforms existing approaches on that matter. From the results, it seems that the method also struggles with the entanglement of pose and appearance in challenging cases, e.g. the video on LSUN in the supplementary or background and hair on FFHQ. It is unclear if auto-encoder-based methods perform similarly well in simpler cases or if the proposed method really improves the disentanglement.
- The method might require as much tuning as the baseline approaches. As one key limitation of TPS transformation-based approaches the authors list that they are “notoriously difficult to tune”. But training a GAN also requires a lot of tuning for each dataset and can be very unstable, e.g., the proposed method is highly sensitive to the choice of GAN-objective. The proposed method further introduces two hyperparameters (tau and K) which might also require tuning for each dataset. Hence, it is questionable if the proposed method is really easier to tune than existing approaches.
- The results in the paper and supplementary seem curated and no uncurated samples are shown. Therefore, it is difficult to predict if the shown results are cherry-picked.

### Ethics statement:
- Second paragraph: One could still optimize for the latent codes of a real image to manipulate it.
- Third paragraph: The claim that GANs have a low risk of memorizing training images is not supported well enough. If there is work that clearly indicates that training data cannot be recovered from GANs it needs to be cited here, otherwise the claim does not hold.

### Additional questions / comments:
- Did you try to sample keypoints from a pretrained keypoint extractor instead of learning them end-to-end? As this might simplify the task it would be interesting to see if it can further improve your results.
- What exactly makes TPS-based methods so difficult to tune?
- I suggest adding the FID results for all datasets to the main paper, as well as the FID results for the compared baselines on those datasets so that the reader does not need to rely on a few qualitative results only.
- Lorenz et al. also evaluate on Human3.6M and PennAction which are very interesting for keypoint-based editing as they consider more complex scenes than e.g. celebA. Evaluating these datasets would substantially strengthen the claim that the proposed method “can be applied to many different objects and image domains”.
- On the BBC pose videos it looks like the GAN overfits to the appearance codes, because e.g. the head pose changes but keypoints remain at the same position. This is an interesting and reasonable failure case that should be added to the limitations.
- Appendix B: why do you not always use the tuned LatentKeypointGAN? What are the downsides of this architecture? Please add the FID evaluation similar to Table 1. Or is this the ‘keypoints’ row? Then the naming needs to be more clear.
- Adding more and especially uncurated samples on all datasets to the appendix, e.g. latent interpolations of single keypoints would be really useful to get a better intuition on how well your approach works.
- Related work: your approach would be even better motivated if you could state more clearly why it is important to tackle this problem in an *unsupervised* setting. Is it difficult to obtain reliable segmentation masks/landmarks in the settings you consider?
- Intro: Both works (Li, Xu) that list limitations of auto-encoder-based approaches are only arxiv publications in your references which is not particularly strong as an argument. (Li is actually accepted at ICPR though). It would be more convincing to illustrate the limitations (e.g. entanglement) of the approaches here.
- FFHQ also has landmarks available. For completeness, it would be good to report the keypoint detection results on FFHQ as well.
- Table 1: Why is FID without keypoint conditioning better than with?
- Table 2: Why is FID so much worse with additive global style while L1 error gets better? (I think you mention this in the appendix, but it might be good to add it here already)
- How useful is the additional loss on the background? You could add a (qualitative) ablation to show that it is needed.

### Misc:
- Throughout the paper: if you refer to experiments in the appendix, please also put the link to the corresponding section.
- First paragraph sec 3.1 “MLP to respectively generate the”
- Last paragraph sec 3.1 “We show in the evaluation ... ” Please add a link to the respective section in experiments.
- Sec 3.1: Add information that K is a fixed hyperparameter
- Sec 3.3: Please explain spatial adaptive normalization so that the paper is self-contained
- Sec 4.1 celebA: please explain MAFL to be self-contained
- Sec 4.1 celebA: celebA-HQ version missing that is used for sec 4.4, also why do you use celebA-HQ there and not just celebA?
- Sec 4.2 “appendix document” → “appendix”
- Sec 4.4 celebA-HQ: what’s the resolution?
- Table 2: contrastive learned keypoint embeddings are not explained in the method section, not sure what you mean by this
- Appendix B:  “the convolutional generator is translation invariant” this is not trivially true, see “Alias-free Generative Adversarial Networks (StyleGAN3)” by Karras et al.
- Figure 9: (top) (bottom) should be (left) (right)

**Summary Of The Paper:**

The work proposes a novel method for **locally controllable image synthesis via keypoints**. In contrast to existing auto-encoder frameworks, this work considers a GAN-based method to overcome the following limitations:
- careful tuning of existing keypoint-based methods
- better disentanglement of pose and appearance

The method operates in an **unsupervised** setting, i.e. training with rgb-images only.

**Summary Of The Review:**

The paper shows **impressive results** for keypoint-based editing of synthesized images. However, there are **too few qualitative and quantitative comparisons to existing approaches** which makes it difficult to assess the quality of the proposed method in practice. In particular, it is not clear if the proposed method indeed outperforms existing approaches wrt. generalization, i.e. being easier to tune, and disentanglement of pose and appearance. Therefore, in its current state, I lean towards rejecting the paper.


**Post-Rebuttal**:
After the rebuttal, I lean towards accepting the paper as image fidelity improves over the baselines. However, my concerns regarding better performance wrt disentanglement and easier tuning than the baselines remain.

---

> ### Author Response · Authors · 2021-11-23
> **Rebuttal - Jmzu - Part 1 (Setting and generality)**
>
> __More motivation for using the unsupervised setting__
>
> For the face and human dataset, it is not difficult to obtain keypoints and masks, however, already keypoint sets with varying granularity are unavailable. Besides this, one needs to consider the ethics (or licensing) problems when one wants to use a public dataset or collect a dataset, which makes this method also commercially interesting. We test our model on these datasets because they are well-established open benchmarks. LSUN Bedroom is our prime example of a domain where no keypoint annotations exist.
>
> __Experimental section does not validate disentanglement? Quantitative evaluation?__
>
> Quantifying spatial disentanglement is difficult. As a first surrogate, we quantify improved consistency of the keypoint positions (which evaluates the disentanglement of position part positions from each other and from others). On the appearance side, we used scripted editing operations to disclose whether parts can be exchanged and modified independently. Since image quality after editing is without ground truth difficult to quantify computationally. To this end, we conducted an extensive user study in Section 4.8. Part of it is a comparison of how realistic an image looks after assigning appearances from one image to the location at another. Our subjects rated our editing results much higher than all the tested competitors, which demonstrates the better appearance disentanglement. In particular, we compare against auto encoder-based approaches which validate our main claim of achieving a better disentanglement. We compare against supervised and unsupervised approaches.
>
> __Representative qualitative results?__
>
> The examples in the user study are randomly generated, see Appendix D. Some curation was necessary to match faces from different GAN methods that are roughly in the same pose and of similar complexity since all generated images are random. Otherwise, by comparing frontal and side views, users would choose, e.g., based on the scale in the image instead of the editing techniques. Please inspect the newly provided entirely randomized results (supplemental document, explained in the general comments to all reviewers).
>
> __Parameter sensitivity?__
>
> Our GAN method is not difficult to tune. Although it is sensitive to the type of GAN-objective functions, we did not need to tune the losses for individual datasets. We have added ablation tests on tau and the number of keypoints in Appendix F2 to show a large range of working parameter choices.
>
> __Ethics statement__
>
> While Brock et al. [A] and Karras et al. [B] argue that GANs do not memorize training datasets, recently Feng [C] empirically showed that a GANs ability to memorize training datasets depends on the complexity of the training datasets. We already acknowledge that one could train an encoder to our latent space. Hence, we acknowledge that there is a possible risk that our GAN can be abused. We added this discussion to the ethics section.
>
> __How useful is the additional loss on the background?__
>
> The background loss is not essential but stabilizes the keypoints. Without it, the keypoints still work but some may move onto the background. We added an ablation test to show the effectiveness of the background loss in Appendix F. Please see the discussion with TTKS.
>
> __Adding more and especially uncurated samples on all datasets to the appendix__
>
> Done. These are indeed helpful to validate the representative value of our main results.
>
> [A] Andrew Brock, and Jeff Donahue, and Karen Simonyan. Large Scale GAN Training for High Fidelity Natural Image Synthesis. In International Conference on Learning Representations, 2019.
>
> [B] Tero Karras, Timo Aila, Samuli Laine, and Jaakko Lehtinen. Progressive growing of gans for im- proved quality, stability, and variation. In International Conference on Learning Representations, 2018.
>
> [C] Qianli Feng, Chenqi Guo, Fabian Benitez-Quiroz, and Aleix Martinez. When do GANs replicate? On the choice of dataset size. In International Conference on Computer Vision, 2021.

---

> > ### Author Response · Authors · 2021-11-23
> > **Rebuttal - Jmzu - Part 2 (Autoencoder and TPS)**
> >
> > __Struggling on LSUN bedroom and no auto-encoder methods compared here?__
> >
> > LSUN bedroom is a challenging dataset because it does not show a single object but multiple ones scattered around the image, which violates the assumptions (e.g., prior on the connection of parts) of other keypoint-based approaches. We could not meaningfully compare on the LSUN bedroom as no existing TPS-based auto-encoder was applied to it in the literatur. We now trained a refined version of the code [https://github.com/theRealSuperMario/unsupervised-disentangling/tree/reproducing_baselines] of Lorenz et al. 2019 on LSUN and show the results in the Appendix C, Figure 10. Keypoints are scattered around the image with mostly static position and without a meaningful relation to parts. We believe that this dataset violates the assumption of smooth deformation of parts via TPS. In comparison, we used our method to sample 100 images as pose the source and independently 100 appearance source images and combined them to yield a more complete picture of editing quality. In total, there are 31% of images of low quality that do not lend themselves for editing. Figure 41 in the supplemental shows that these are easy to identify during editing. For all other images, editing operations are of good quality.  Note that about 10% of the dataset images are of low quality too, showing just the bedding, other closeups, or advertisements. Hence, some selection during editing is inevitable.
> >
> > __Evaluation on Human3.6M and PennAction?__
> >
> > We tried applying it to PennAction in the short period of the rebuttal. Without adjustment of hyperparameters, keypoints on PennAction are mostly scattered on the background regions with only a few on the person. It seems that TPS models, e.g., Lorenz et al., apply better to such articulated human motion while ours generalizes better to less structured datasets that contain large shape variation, e.g, bedroom (see prev. answer). We expect similar results on Human3.6M. It would be natural to try to combine the advantages of both in future work.
> >
> > __What makes TPS-based methods difficult to train?__
> >
> > The range of the randomized TPS parameters must be able to explain the transformations that exist in the dataset. This is difficult since each dataset has different distributions of object transformations. On the empirical side, we tried re-implementing TPS models of Lorenz et al. for a long time and could not reproduce their excellent results. Reproducibility is also an issue on associated GitHub pages [https://github.com/CompVis/unsupervised-disentangling/issues/3]. For TPS, a typical number of parameters is 6 to 20. By contrast, our method only requires choosing the size tau and number of keypoints K, which is more intuitive and easier to determine.
> >
> > __Illustration of the limitations of auto-encoder based methods__
> >
> > We have shown their editing results in our survey, which clearly shows that our model outperforms theirs in image quality. The training of the Lorenz et al 2019 re-implementation on the LSUN bedroom (Appendix C, Figure 10) further supports this claim.

---

> > > ### Author Response · Authors · 2021-11-23
> > > **Rebuttal - Jmzu - Part 2 (Details)**
> > >
> > > __Have you tried to sample from a pre-trained keypoint extractor?__
> > > Using a supervised keypoint detector would defy our goal of an unsupervised method. It is an interesting avenue for future work to use our unsupervised keypoint detector to drive generated models with a real source video.
> > >
> > > __More FID results?__
> > > The main paper already reports FID for those where the most related methods report. We added additional FID scores in Appendix C for all other datasets to aid future comparisons. For autoencoders, it is not common to use FID. Thanks for the suggestion.
> > >
> > > __Overfitting on BBCPose?__
> > > We observed that our model chooses to explain head rotation with an appearance change instead of pose change. It is indeed an interesting failure case, and we will add that to our limitations. Rotation is not explicitly encoded and is an additional direction for future work.
> > >
> > > __Why not use tuned-LatentKeypointGAN for editing?__
> > > While the tuned-LatentKeypointGAN gives higher accuracy on keypoints, we do not observe significant improvements in editing. Furthermore, it has a more complicated architecture, is computationally more expensive, and has a larger memory footprint during training. Hence, we opted for the simpler model for those tasks where it suffices. We added the FID for tuned LatentKeypointGAN in Appendix B.
> > >
> > > __Keypoint detection results on FFHQ?__
> > > We did not manage in time but could provide results during the discussion phase.
> > >
> > > __Why is the FID without keypoint conditioning better than with keypoint conditioning?__
> > > It is difficult to disentangle an object into parts. Therefore, if we remove this constraint, networks can generate images more freely which results in better image quality, though without being able to edit or enabling the generation of pseudo labels for detection.
> > >
> > > __Why is FID worse with additive global style while L1 error gets better?__
> > > These measure very different quantities (image quality vs. Positional conditioning) and as the previous question and answer reveal, image quality usually degrades with additional constraints. In particular, the additive keypoint embedding is not powerful enough to fully capture the variations of the key parts. As a result, the background controls the foreground appearance. The dynamic embedding w_global now works as noise to prevent the network from hiding foreground appearance information in keypoint locations. As a result of good disentanglement of appearance and keypoint location, the keypoint detection accuracy slightly increases. However, again, due to the less powerful expression of additive keypoint embedding, FID decreases. For more details, please refer to Appendix F.
> > >
> > > __Missing SPADE explanation?__
> > > We included the explanation in the Appendix, due to space constraints. We added a link in the main paper to the location in the appendix.
> > >
> > > __Why use CelebA-HQ instead of just CelebA?__
> > > We clarified in the revision that we use CelebA- HQ of resolution 256x256 to compare fairly with SEAN that uses the same resolution, and otherwise use CelebA of resolution 128x128.
> > >
> > > __What are the contrastive learned keypoint embeddings?__
> > > Here, we deploy a contrastive loss to disentangle encodings. Since contrastive learning is a common tool, we tested the contrastive learned keypoint embeddings in appendix F1 to demonstrate the superior effect of our multiplicative embedding.
> > >
> > > __Convolutional generators are translation invariant?__
> > > Convolutions are translation invariant. However, convolution with zero padding will violate the translation equivariance of the feature maps. This is exactly why we design the tuned LatentKeypointGAN to avoid such leaks.
> > >
> > > __Misc.__ Thank you for the detailed review. We added and modified the lines you noted.

---

> > > > ### Comment · Reviewer_Jmzu · 2021-11-30
> > > > **Response to Rebuttal**
> > > >
> > > > Thank you for addressing my concerns in such detail and providing additional results and insights.
> > > >
> > > > Regarding my major concerns:
> > > > 1. Missing support that existing methods struggle more with disentanglement
> > > > While I think that higher user ratings do not quantify disentanglement (but rather image fidelity, similar to FID), I understand that disentanglement is difficult to measure. However, I still think that qualitative comparisons regarding disentanglement would be important to support this claim.
> > > > 2. Tuning might be as difficult as tuning baselines
> > > > Thank you for adding the ablation on tau and K. My concern partially remains though, because tuning the method to PennAction seemed to be difficult and hyperparameter tuning seems to be crucial.
> > > > 3. Representative results
> > > > Thank you for adding so many uncurated samples, this is very helpful.
> > > >
> > > > Considering that some important concerns remain but most of my concerns were addressed successfully by the authors, I change my rating to a 6.

---

> > > > > ### Author Response · Authors · 2021-11-30
> > > > > **Note**
> > > > >
> > > > > 1. In comparison to methods that have lower image quality, that concern is valid. However, please note that the strongest competitor, SEAN (Zhu et al., 2020b), has a better image quality (FID score, see main paper Table 1) before editing. Their images are indeed as good or better as ours. Only after the editing operation, our results are rated higher (see main paper Section 4.8 and Appendix Table 4). The degradation is because artifacts are introduced when exchanging part embeddings, i.e., parts are entangled with other parts or the positioning of parts.
> > > > >
> > > > > We argue that image quality after editing measures an aspect of disentanglement and this is a fair comparison if the competing method scores the same or better before the editing operation. Therefore, our method suffering from fewer artifacts as well as being preferred on the disentanglement questions validates the claim of a better disentanglement in direct comparison to SEAN.

---

### Official Review · Reviewer_TTKS · 2021-10-28

**Correctness:** 3
**Technical Novelty And Significance:** 3
**Empirical Novelty And Significance:** 2
**Recommendation:** 5
**Confidence:** 4

**Main Review:**

I think overall this idea is interesting and straighforward, where controlling the synthesis process via keypoints is pretty useful in practice.
However, I think many related works that provide structural and local edits of GAN synthesis process are missed, making the experiments not very comprehensive.

Major concerns:
1. Many related works are not included and discussed:

[a] Kim, H., Choi, Y., Kim, J., Yoo, S., & Uh, Y. (2021). Exploiting Spatial Dimensions of Latent in GAN for Real-Time Image Editing. In Proceedings of the IEEE/CVF Conference on Computer Vision and Pattern Recognition (pp. 852-861).

[b] Kwon, G., & Ye, J. C. (2021). Diagonal Attention and Style-based GAN for Content-Style Disentanglement in Image Generation and Translation. ICCV 2021

[c] Edo Collins, Raja Bala, Bob Price, and Sabine Susstrunk. Editing in style: Uncovering the local semantics of gans. In CVPR, 2020

[d]  Yazeed Alharbi and Peter Wonka. Disentangled image generation through structured noise injection. In CVPR, 2020.

[e]  Peihao Zhu, Rameen Abdal, Yipeng Qin, and Peter Wonka. Sean: Image synthesis with semantic region-adaptive normalization. In CVPR, 2020.

[f]  Fangneng Zhan, Hongyuan Zhu, and Shijian Lu. Spatial fusion gan for image synthesis. In CVPR, 2019.

2. I consider latentkeypointGAN as a type of unconditional gan, thus it should not be compared to Pix2PixHD/SPADE/SEAN but recent works that also incorporate structural latent space for unconditional gans, e.g. the ones mentioned in point 1.

3. Another question is that a background loss is adopted to disentangle background semantics from foreground semantics, will the semantics of these k keypoints be entangled with each other? Did you observe more than one keypoints control the same region (i.e. some kind of collapose) during training? If so, how should we resolve this? And if not, why?

4. K is a pre-defined value? What do you mean by adding a keypoint in Figure 1? How should we choose the value of K? There is no ablation study about this.

**Summary Of The Paper:**

This paper proposes to use a set of structured keypoints as the intermediate representation for image generation, so that ideally each keypoint with control the location and semantics of a certain part. For each generated image, three noise vectors will be sampled from a gaussian prior distribution, which respectively estimates the locations of k keypoints, the global image style, and the semantics of background. Training is achieved by the standard GAN loss plus a background loss that disentangles background semantics from foreground semantics.

**Summary Of The Review:**

I think the idea is good. But related works are not discussed thouroughly. I also have concerns in terms of the baselines and ablation studies in experiments.

---

> ### Author Response · Authors · 2021-11-23
> **Review TTKS**
>
> Thank you for your time and detailed feedback. Please don’t miss our general notes including hundreds of new qualitative results that validate those fewer ones in the main paper.
>
> __Missed references__
>
> Thank you for pointing us to additional related work, we were not aware of [a-d,f]. We already compare against [e] (Zhu et al., 2020b) in Table 1, Figure 9, Appendix D, and we also added more comparison in Figure 26-32 in our new supplementary document. [b] was just published at ICCV’21, a week after our submission.
>
> To better put our method in context, we re-wrote our related work section and included [a-d] in the revised version. In sum, all these methods differ significantly from our goal and contributions in that we introduce a way to condition on keypoints while these methods condition on less structured pixel maps, which requires manual pixel-level selection of the region that should be edited. Our keypoint-based solution requires less manual intervention and enables new editing operations, such as changing the relative position of parts.
>
> Detailed discussion:
>  * Kim et al. (2021) [a] Their model enables editing of parts by modifying a spatially variable code mapping. However, their editing requires drawing pixel-level regions at source and target. Moreover, editing is not always localized, e.g., in the second row in their Figure 4, the head pose changes by modifying the areas that are in the background. On the contrary, our model edits the pose by changing the keypoints on the foreground, which is more intuitive, justifying our new keypoint-based method.
>  * Kwon & Ye (2021) [b] They use an attention map to control the spatial structure of the image, thereby enabling local shape editing and object pose changes by modifying attention maps. However, unlike our work and those we compare against (e.g., SEAN), they do not show that their model can change the appearance of a specific part or add and remove some parts. Besides, this paper came out in ICCV 2021, which is a week after our submission.
>  * Collins et al. (2020) [c] They use spherical k-means clustering to cluster feature maps into several groups and enable localized part appearance editing by swapping the chosen part of the feature maps. However, they require manual selection of suitable clusters out of hundreds of feature maps (see their Section 3.1) and do not enable moving individual parts spatially since the support of clusters remains unchanged. Furthermore, Figure 3 in their paper shows that the part appearance is often not completely swapped. For example, the lip color does not change when it is swapped from male to female, and the nose can hardly be recognized as “exchanged” in all cases. Our core contribution of conditioning on keypoints overcomes both of these limitations.
>  * Alharbi & Wonka (2020] [d] generates images via a stylemap with spatially independent noise factors. In this way, they can edit a local image part by exchanging the respective stylemap. However, this requires manual selection of pixel-level mask regions which makes editing more cumbersome and precludes fully automatic editing as demonstrated in our main paper and the new supplemental examples. Moreover, they cannot swap the part appearance when original and reference images have different poses and target semantic sizes. See their section E.7. Failure cases for more details. On the contrary, with our keypoint approach, we can not only swap appearance between different poses but also change the pose using keypoints. Our method uses similar principles of disentangling the noise of different image regions, but does so at the level of keypoints/parts, which includes disentangling position from appearance, which requires the whole keypoint-based generator architecture we propose.
> * Zhu et al. (2020b) [e] We already compare to SEAN qualitatively and quantitatively in Section 4.4 and Appendix C D, and we provide further 105 editing comparison in the new supplementary document, Figure 26-32.
> Zhan & Zhu [f] They use a cycle consistency with a spatial transformation network to fuse a foreground object on top of a background reference. They can add a part but only if separate datasets with and without that part are available (e.g., with and without glasses) Thereby, their work requires stronger supervision, and is very different to ours (as well as the above methods).
>
> Compared with all the above methods, our keypoint-based GAN provides explicit sparse representations on parts, so that the editing is more intuitive (no area of effect painting), and it can add, remove, and re-position the parts.
>
> On the technical side, obtaining such keypoint conditioning requires fundamentally different ways to process feature maps. Besides, our model decides the appearance information at the beginning of the network, while the above methods modify on the middle-layer feature maps. As a result, we do not need to manually choose in which layer to do the editing.

---

> > ### Author Response · Authors · 2021-11-23
> > **Rebuttal - TTKS - Part 2**
> >
> > __Choice of baselines?__
> >
> > Our LatentKeypointGAN is an unconditional GAN, which is indeed different from Pix2PixHD/SPADE/SEAN. Note that we already compare to Zhang et al. which are unconditional as well. In addition, we include comparisons to SEAN because we share similar functionality and it is important to evaluate the differences between supervised and unsupervised editing. The editing ability of both SEAN and our model includes
> > a) changing all parts appearance and pose (globality)
> > b) changing the appearance of some parts (locality)
> > c) re-positioning, adding, and removing parts (strong disentanglement)
> > While the methods listed above can perform a), b), or both, none of them shows their ability to do c). This editing is one of our main contributions. Both, our model and SEAN, can achieve simple and intuitive editing while the current unconditional GANs manipulate the middle-layer feature maps to achieve local editing, which requires manually specifying the region of effect. In addition, neither of these methods attempted to estimate keypoints, which is an additional advantage of our method with its own applications.
> >
> > __Can keypoints be entangled?__
> >
> > The background requires global context and, therefore, behaves much differently from the foreground. For the keypoint, we have our keypoint generator that disentangles them based on independently learned embeddings. As mentioned in the limitation section, in rare cases the opposite of being too disentangled happens, e.g., where two eyes receive an independent eye color or differently shaped glasses. Entanglement between keypoints can happen when the number of keypoints is large. For instance, with >12 keypoints on the face datasets, there are two or three keypoints controlling the mouth---but this is expected behavior. We also observe that a larger keypoint size separates keypoints more. Dependent on the application case, one can always choose a model with fewer keypoints and a larger keypoint size to control a larger part or use more keypoints with a smaller size to control finer areas. Please refer to our new Appendix F for more details. We further analyzed the keypoint disentanglement in our user study in Section 4.8, in terms of part disentanglement and pose/appearance disentanglement. Our model outperforms Zhang et al. and SEAN. We provide more details in our discussion with reviewer Jmzu.
> >
> > __Is K predefined? How do you choose and how do you remove and add?__
> >
> > Yes, the number of keypoints K is a pre-defined value at training time. While we already provide an ablation study in Appendix F2, we added more rigorous experiments in Appendix F2.
> > As points are cast to feature maps as the concatenation of heatmaps, one can add or remove an arbitrary number of keypoints and associated embeddings. We will clarify that by removing a keypoint in Figure 1, we mean removing its peak in the Gaussian heatmap. By adding a keypoint in Figure 1, we mean adding another peak with associated embedding in the Gaussian heatmap of the chosen keypoint. In the first row (face), we first move all keypoints slightly to the left, and then add all the keypoints to the right. In the second row, we add another keypoint of the drawing (the one being duplicated is marked as green in the second column) and put it on the marked location. It was a toy experiment that surprised us how well the model generalizes with a different number of removed or added keypoints.

---

> > > ### Comment · Reviewer_TTKS · 2021-11-29
> > > **Thank you for the response**
> > >
> > > Hi at first I'd like to thank authors for their effort for making the responses to my concerns. Point 3 and 4 of my concerns have been addressed.
> > >
> > > For point 1&2, I'm not indicating the novelty of this paper is limited given my mentioned related works, so I agree with authors on the per-paper differences between this paper and them.
> > >
> > > I just think this work is an unconditional GAN that provides better latent controllability. Considering this, only comparing to three conditional baselines under the metric FID is not sufficient to fully validate the claim of the proposed method.
> > >
> > > And comparing to a bunch of unsupervised landmark detection methods (Table 3 and Table 4) is a good plus but not a must for this paper in my opinion, since the focus is controlling gans.
> > >
> > > Therefore, I still think authors should compare to unconditional gans that also aim for better controllability, in terms of commonly used metrics such as PPL and LPIPS. Otherwise, this work should be better described as an unsupervised keypoint learning method via adversarial learning.

---

> > > > ### Author Response · Authors · 2021-11-29
> > > > **Quick reply**
> > > >
> > > > Thank you for the additional comments. The requested LPIPS is a metric to compute distances between two image patches, i.e., the reconstructed image to a reference image. However, in our GAN setup, there is no reference image to compare to. This is only possible for methods that condition on the target image with an encoder. For instance, [a] only uses LPIPS to evaluate image reconstruction accuracy, not editing. The FID score we use is essentially a perceptual distance in the distributional sense, which is the most widely used metric for GANs.
> > > >
> > > > PPL has been used to analyze the smoothness in StyleGANs and applying it to two random samples has later been found to be a good image quality metric. It has also been used by [b,d] to show disentanglement between style codes, but not the disentanglement of positional and appearance codes. We are unsure whether PPL would meaningfully generalize between different network architectures. Their comparison is specific to StyleGAN and its variants in that a certain layer is sampled and interpolated. With a different architecture, the sampled points/images could be closer or further away in latent/image space, leading to smaller or higher PPL values. We will nevertheless attempt to get numbers by evaluating our existing models ASAP. Even if not comparable to other models, PPL could at least strengthen our ablation study to additionally quantify the importance of our contributions.
> > > >
> > > > Besides following this new suggestion, we do not see a meaningful way to quantitatively compare local disentanglement against [a-d]. Qualitative comparisons are already possible by comparing our results to the images published in their papers, e.g., the Figure 3 in [c] that we refer to in our rebuttal. These clearly show the advantages of our approach.
> > > >
> > > > Our core contribution is to enable new editing capabilities, such as re-positioning of parts; which has not been demonstrated by the existing GAN approaches (except SEAN). That is why we believe our current comparison to SEAN and the autoencoder-based solutions, which have this capability, is the most meaningful evaluation possible.
> > > >
> > > > ## Edit: Please see the additional evaluation on PPL scores in the post below (hidden behind a link ->)

---

> > > > > ### Author Response · Authors · 2021-11-30
> > > > > **PPL scores - inconclusive**
> > > > >
> > > > > We applied the PPL following the implementation of StyleGAN2 and the LPIPS implementation of https://github.com/richzhang/PerceptualSimilarity. As we suspected, PPL scores seem not transferable across neural network architectures. The PPL when measuring path length in the entire noise input is 2117.0, which is already an order of magnitude larger than for the style GAN variants. We then measured the PPL in different points of the latent space, as done for [b,d]. The table below lists all quantities. The PPL in the disentangled latent spaces reduces; it goes down from 2106.1 in $z_{kp noise}$ to 1471.8 for the latent positions $k$ and from 303.8 in $z_{kp app}$ to 112.3 for $w_{global}$. The latter is in the range of values observed by StyleGAN methods.
> > > > >
> > > > > |  | PPL | Dimensions |
> > > > > | --- | --- | ----------- |
> > > > > | Entire noise space | 2117.0 | 512 |
> > > > > | Entire latent space | 1521.3 | 148 |
> > > > > | Noise space of position  | 2106.1 | 256 |
> > > > > | Latent space of position | 1471.8 | 20 |
> > > > > | Noise space of embedding  | 303.8 | 256 |
> > > > > | Latent space of embedding | 112.3 | 128 |
> > > > >
> > > > > Our PPL numbers are generally higher than those reported for StyleGAN variants. We believe one reason for the discrepancy is the scale of the latent space. Our keypoint space is limited to the range [-1, 1], while the appearance embedding, as well as the latent space for StyleGAN, is unconstrained. However, arc length is sensitive to scale. Mathematically, the formula for arc length in PPL is of the form $L = d(G(w), G(w + \epsilon))^2/\epsilon^2$, with $d^2$ the LPIPS score that is a squared distance, $G$ the generator, $w$ the latent code, and $\epsilon$ a small constant for finite differences computation. E.g., scaling the embedding network output by 10 and undoing the same in the generator G yields a 10-times smaller arc length as the eps is a constant; albeit this scaling does not change the overall function nor the disentanglement. Formally, the scaled PPL is $ L' = d(G(w 10/10), G((w 10 + \epsilon)/10)^2/\epsilon^2 $ $= d(G(w), G((w + \epsilon/10))^2/\epsilon^2$, leading to a 10-times smaller PPL score as the factor $\epsilon/10$ remains (assuming that eps is small and hence d and G linear). The scaling of our latent spaces indeed differs; the std of keypoint location $k$ is a low 0.47 while the embedding std is 4.9. Hence, changes in PPL can only be attributed to disentanglement if other factors are eliminated; an additional normalization would be necessary.
> > > > >
> > > > > However, it is likely not as easy as normalizing to a fixed scale since also the latent space dimension has an influence. Our latent space is much smaller (20 dim for position and 128 for the embedding). In a similar vein to the scaling issue, embedding the same amount of sample points over a larger space in higher dimensions will lead to a flatter embedding structure (lower gradients, what the arc-length computation is a function of https://en.wikipedia.org/wiki/Arc_length). It calls for a separate paper on how to best compute smoothness and disentanglement across network architectures and dimensions or to find a different function to measure disentanglement. In the absence of such, we resorted to a user study and qualitative comparisons.

---

### Official Review · Reviewer_kpkc · 2021-11-02

**Correctness:** 3
**Technical Novelty And Significance:** 3
**Empirical Novelty And Significance:** 3
**Recommendation:** 6
**Confidence:** 4

**Main Review:**

> Authors proposed an interesting idea that can detect keypoints of the object in the unsupervised way and offer the capability of manipulating images based on the detected keypoints.

> Few related works are missing: Authors proposed the interesting idea and implemented it in the effective way. However, I could remember an approach that already provide the interactive capability for GAN by representing its intermediate representation as the segmentation map. I think authors need to add this reference and discuss their difference. Personally, in the sense that the segmentation map is similar; yet more advanced than the 2D heat maps, authors contribution could be significantly reduced by this paper.

[1] Learning Hierarchical Semantic Image Manipulation through Structured Representations, NeurIPS'18.

**Summary Of The Paper:**

Authors proposed the LatentKeypointGAN that detects keypoints and provides the editable capability using the keypoints. The overall training is performed in the unsupervised way without the explicit keypoint supervision. In the aspect that GAN provides the capability of generating the images with the photo-realistic quality while the ability to control attributes are limited, the proposed methods contribute to the related society.

**Summary Of The Review:**

Overall, I am at the borderline for this paper; however towards more accept side. I could move towards the higher score, if authors could rebut my points effectively.

---

> ### Author Response · Authors · 2021-11-23
> **Rebuttal - kpkc**
>
> Thank you for your time and the important pointer to [1]. Please do not miss our general notes including hundreds of new qualitative results that validate those fewer ones in the main paper.
>
> __Relation to “Learning Hierarchical Semantic Image Manipulation through Structured Representations, NeurIPS'18.”?__
>
> The paper [1] shares a similar goal of local image editing, however, in a supervised setting. They proposed a hierarchical model that first infers segmentation masks from bounding boxes and then generates images based on the masks. In this way, they can edit a local part of the image by manipulating a bounding box. However, their approach requires ground-truth semantic label maps at training time. We integrated the suggested paper into the discussion of related supervised approaches (marked in blue in the related work section). Such pixel-level annotation can be very expensive and is not available for every domain. That is why we target the unsupervised case. Nevertheless, these approaches are relevant since we build upon them by using similar network architectures (discussed in Section 2, Conditional image synthesis). Our goal and contributions are reaching similar editing capabilities without any labels at training and test time. This enables editing and keypoint detection in new domains where no labels exist, such as for the bedroom dataset.
>
> Note that we indeed use labelled datasets for testing our algorithm quantitatively. However, no labels are used at training time. At test time, labels let us compute accuracy numbers, such as to evaluate our localization accuracy and to quantify the gap between unsupervised and supervised approaches.

---

> > ### Comment · Reviewer_kpkc · 2021-11-29
> > **Thank you for the response.**
> >
> > I initially offered the borderline accept for this paper. I agree that the unsupervised characteristic of this paper provides some difference to previous literature; however I still think that this seems not very huge contribution to the community yet. So, I'd like to stay in the borderline accept.

---

### Author Response · Authors · 2021-11-23
**General Points**

We thank all the reviewers and chairs for their time and detailed feedback. We focus our rebuttal on answering open questions and supporting prior experiments with a new supplemental document of hundreds of qualitative results and include the requested quantitative experiments in the refined appendix. Below are answers to the two most important questions. The other questions are addressed in the individual replies.

__Are results and comparisons representative?__

Our figures in the main paper are representative. To show this, we updated the supplemental with literally hundreds of additional examples that are generated without any manual intervention, for the same editing operations shown in the paper. We also included additional examples from SEAN to compare more rigorously. The file can be found in “examples.pdf” in the supplemental. It includes an explanation of how we automatically generate the following editing examples:

Figure 1-7: editing on FFHQ;
Figure 8-9: editing by removing keypoints one by one;
Figure 10-21: individual keypoint embedding interpolation;
Figure 22: editing on BBCPose;
Figure 23-24: editing on LSUN Bedroom;
Figure 25: Failure cases on LSUN Bedroom;
Figure 26-32: Editing Comparison with SEAN (Zhu et al., 2020b)

__What is the influence of the number of keypoints and their size tau?__

On top of the ablation tests on tau and the number of keypoints that we already had in the Appendix F2, we added more tests on different combinations of tau and the number of keypoints in the new Appendix F2 (at the end of the updated submission, new paragraphs marked in blue) to analyze their interplay. The approach is robust to different combinations of these two hyperparameters.

---

### Decision · Program_Chairs · 2022-01-20

**Decision:**

Reject

**Comment:**

The paper proposes an unconditional GAN that learns a set of structured keypoints as the intermediate representation. It was shown that these learned keypoints may be used to control the image synthesis output. The paper received a mixed rating before the rebuttal, with one reviewer rating the paper marginally above the bar and three reviewers rating it marginally below the bar. While a couple of reviewers commented that the keypoint idea was interesting, several concerns were raised, including the seemingly challenging tuning requirement and the usability of the proposed method. Several missing related works were also pointed out. The rebuttal addressed some of the raised concerns but not fully. While Reviewer Jmzu raised the score from marginally below the bar to marginally above the bar, Jmzu still expressed concerns about the quality of the paper. Reviewer kpkc kept marginally above the bar rating but was not impressed with the contribution. Consolidating the reviews and rebuttals, the meta-reviewer found the raised concerns valid and would not recommend acceptance of the paper. The authors are encouraged to incorporate the feedback to strengthen the contribution.